# FUSING MODELS WITH COMPLEMENTARY EXPERTISE

**Hongyi Wang,**♠ **Felipe Maia Polo,**♦ **Yuekai Sun,**♦
**Souvik Kundu,**▲ **Eric P. Xing,**★♠¶ **Mikhail Yurochkin**♣
♠ Carnegie Mellon University   ♦ University of Michigan   ▲ Intel Labs
★ MBZUAI   ¶ Petuum, Inc.   ♣ MIT-IBM Watson AI Lab

## ABSTRACT

Training AI models that generalize across tasks and domains has long been among the open problems driving AI research. The emergence of Foundation Models made it easier to obtain expert models for a given task, but the heterogeneity of data that may be encountered at test time often means that any single expert is insufficient. We consider the *Fusion of Experts (FoE)* problem of fusing outputs of expert models with *complementary* knowledge of the data distribution and formulate it as an instance of supervised learning. Our method is applicable to both discriminative and generative tasks and leads to significant performance improvements in image and text classification, text summarization, multiple-choice QA, and automatic evaluation of generated text. We also extend our method to the "frugal" setting where it is desired to reduce the number of expert model evaluations at test time. Our implementation is publicly available at https://github.com/hwang595/FoE-ICLR2024.

## 1 INTRODUCTION

Traditional training of ML/AI models, *i.e.*, empirical risk minimization, allows obtaining experts specialized for a particular task and domain. However, in practice, the distribution of the test data often differs leading to significant performance degradation (Koh et al., 2020; Santurkar et al., 2020). The emergence of Foundation Models (Bommasani et al., 2021) allows obtaining expert models with simple fine-tuning on a handful of data, but such expert models still face generalization issues (Kumar et al., 2022; Jang et al., 2023; Cheng et al., 2023). In the case of Large Language Models (LLMs), although most advanced commercial models can perform well on a range of tasks, they are closed-source, expensive to use, and can be outperformed by smaller specialized models (Hsieh et al., 2023; Gunasekar et al., 2023; Jang et al., 2023; Fu et al., 2023). In other words, training expert models have become extremely effective, while generalization with a single model remains challenging. In this paper, we aim to develop methods for combining the strengths of models with *complementary* expertise to push their collective generalization capabilities.

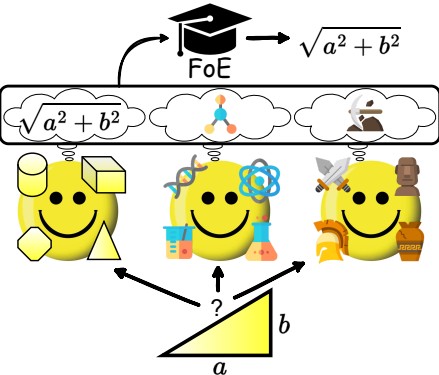

Figure 1: Three experts with complementary expertise (geometry, natural science, and history) process an input question on the Pythagorean theorem. They each output responses that are processed by a Fusion of Experts (FoE) model to arrive at a final output. Note that only one expert is capable of producing a high-quality output, thus ensembling the experts is likely to perform poorly.

Combining experts is a long-standing and successful practice in ML: classical approaches like ensembling (Dietterich, 2000; Ganaie et al., 2022) or Mixture of Experts (MoE) (Jacobs et al., 1991; Jordan & Jacobs, 1994; Masoudnia & Ebrahimpour, 2014) often lead to performance improvements. These techniques typically consider experts trained and tested on the same data distribution, although improvements in terms of out-of-distribution (OOD) generalization have also been observed (Lakshminarayanan et al., 2017; Shen et al., 2021). In contrast, in this paper, we revisit the practice of combining experts in a new setting where we are given pre-trained experts with *complementary* expertise, and our emphasis is on generalization to test data distributions where none of the experts perform well individually. See Figure 1 for an illustration.

More specifically, we consider a problem where the data distribution is comprised of $K$ domains, and we have access to $K$ expert models, one for each of the domains. We do not make any assumptions

about how experts were trained to make it compatible with modern practices of fine-tuning Foundation Models to obtain high-quality experts. At test time, the data is sampled from the mixture of the $K$ domains, *i.e.*, it contains data from every domain, and thus any individual expert will be sub-optimal. Our goal is to train a model using expert outputs that produces final predictions or generations either by choosing one of the experts or by combining their outputs, for a given input data point. We refer to such models as the *Fusion of Experts (FoE)* models. In our experiments, we consider tasks such as image/text classification, text generation, and automatic evaluation of generated summaries (Mani, 2001). Finally, recognizing that obtaining the outputs of every expert can be expensive, we consider the "frugal" setting (Chen et al., 2020) and propose an extension of our method to reduce the number of expert evaluations. Our contributions are summarized below:

1. We formulate the *Fusion of Experts (FoE)* problem of fusing outputs of models with complementary expertise and cast it as a supervised learning problem. Our approach is applicable to both discriminative and generative use cases.

2. We further extend the problem to present *Frugal Fusion of Experts (FrugalFoE)*. In specific, we extend our formulation by casting it as a graph shortest path problem that allows us to efficiently perform expert fusion while only evaluating a subset of experts at test time.

3. We demonstrate the efficacy of our method through extensive experimental evaluations on image classification with standard deep learning methods, text classification, summarization, and question answering using Large Language Models (LLMs), and automatic evaluation of generated summaries. Our proposed fusion method can greatly improve the performance of individual experts, while also reducing the number of expert evaluations at test time.

## 2 RELATED WORK

**Ensemble learning.** Ensemble learning combines several individual models (*e.g.*, either model outputs or model weights directly) to obtain better performance (Breiman, 1996a;b; Ganaie et al., 2022), which has also been justified by theoretical analysis (Hansen & Salamon, 1990; Krogh & Vedelsby, 1994). Classical ensemble methods include bagging (bootstrap aggregating), boosting, and stacking (model blending) (Breiman, 1996a; Freund et al., 1996; Friedman, 2001; Wolpert, 1992). Negative correlation learning, which encourages ensemble models to learn diverse aspects from the training data, has also been widely used for deep ensemble methods (Liu & Yao, 1999; Shi et al., 2018; Zhang et al., 2019a; Buschjäger et al., 2020). Especially in deep neural networks, the ensemble effect can be introduced implicitly by various methods, *e.g.*, Dropout (Srivastava et al., 2014). Most works in this group implicitly assume that models in the ensemble have *similar* expertise, thus it is beneficial to aggregate their predictions. To combine models with *complementary* expertise, averaging their outputs might be detrimental due to most of them being not suitable for an input.

**Mixture of experts (MoE).** The basic idea of MoE is a learned weighted ensemble among "expert models" where the expert(s) choice is made via a "gating mechanism", *i.e.*, that controls the expert selection for a certain inference query/instance or the weight coefficients to combine various experts (Jacobs et al., 1991; Jordan & Jacobs, 1994; Masoudnia & Ebrahimpour, 2014). The idea of MoE has recently been extended to LLMs where multiple MLP experts are integrated after each multi-head self-attention in the Transformer encoder/decoder blocks (Fedus et al., 2022; Chowdhery et al., 2022). The use of MoE in LLMs has been demonstrated to effectively scale the model sizes up further without costing proportional increase in the computation complexity, as the computation in MoE is effectively sparse. The majority of works in this group are designed for the joint training of "experts" and the gating/aggregation module. In our problem setting, the experts are pre-trained on their respective domains and serve as a starting point for the Fusion of Experts.

**Federated/collaborative Learning.** Federated/collaborative learning allows various clients/agents to jointly learn using their own (mostly private) local data (Kairouz et al., 2021; Wang et al., 2021). During the federated learning, participating clients conduct in-situ learning and computing before their locally learned models are communicated to a central server for model aggregation or fusion (McMahan et al., 2017). Common federated model fusion/aggregation methods include various averaging schemes, ensemble-based schemes, and MoE type of gating mechanisms (McMahan et al., 2017; Wang et al., 2020b; Li & Wang, 2019; Lin et al., 2020). Our work can be seen as a special case of federated learning where clients train their own models locally and share it with the central server for training the FoE model to aggregate them.

**Combining pre-trained models.** One common way a practitioner can interact with a pre-trained model is via an API. Such models typically vary in performance and cost. FrugalML (Chen et al.,

2020) aims to maximize the usage of the cheapest API on "easy" inputs, while only querying the more expensive ones when needed. This work has also been extended to LLMs (Chen et al., 2023b). In our setting, the expert models have similar costs and complementary expertise, i.e. no expert is better than any other across all of the data distributions. Finally, (Ravaut et al., 2022; Jiang et al., 2023) train auxiliary LLMs to combine generations of general (non-expert) LLMs, however, do not consider the "frugal selection of experts".

## 3 Fusing models with complementary expertise

The real data distributions are often diverse and challenging to represent with a single dataset and/or mimicked by a single AI model (Santurkar et al., 2020; Koh et al., 2020). For example, in image recognition, different geographic regions has different patterns such as colors of traditional attires (Atwood et al., 2020). In text summarization, models are typically fine-tuned for specific domains such as news articles from a particular publisher, e.g., CNN (Lewis et al., 2019), or WikiHow articles (Cohen et al., 2021). To model such distributions, we consider a mixture model with $K$ components:

$$D = \sum_{k=1}^{K} \alpha_k D_k, \ \sum_k \alpha_k = 1, \ \alpha_k > 0 \ \forall \, k, \tag{3.1}$$

where each $D_k$ represents a subdomain of $D$ such as images from a specific geographic location. Our goal is to train an AI system that can perform well on $D$ leveraging the recent advances in Foundation Models that allow us to obtain powerful models (experts) for any given $D_k$ even when training data from $D_k$ is scarce (e.g., via few-shot learning) (Brown et al., 2020). Such experts alone typically struggle to perform well on $D$ due to the performance drop on any domain apart from the one they were fine-tuned on (Jang et al., 2023). In this work, we explore strategies for fusing different expert outputs to improve performance on $D$. Thus, the input to our problem is a set of models $\{f_k : \mathcal{X} \to \mathcal{Y}\}_{k=1}^{K}$ and some amount of validation data $\{x_i^k \in \mathcal{X}, y_i^k \in \mathcal{Y}\}_{i=1}^{n_k}$ from every domain $k = 1, \ldots, K$, where $\mathcal{X}$ and $\mathcal{Y}$ are input and output spaces correspondingly. We consider supervised and generative scenarios, i.e., $\mathcal{Y}$ can be numerical, categorical, or representative natural language.

### 3.1 Fusion of classification experts

Let us consider a supervised learning scenario with $C$ classes, i.e., $\mathcal{Y}$ is the probability simplex $\Delta^{C-1}$. Our goal is to fuse the expert outputs, i.e., to learn a mapping from $\{f_k(x)\}_{k=1}^{K}$ to associated label $y$. We review common measures to combine expert predictions, then present our fusing approach.

The traditional approach to combine expert predictions is to average their outputs, i.e., ensembling: $\frac{1}{K} \sum_k f_k(x)$. Another popular choice is to use the most confident model (Pearce et al., 2021; Chen et al., 2023a), i.e., predict with the model that outputs the largest probability for a class: $f_{k^*}(x), \ k^* = \arg\max_k \max_c \ [f_k(x)]_c$. Model confidence is often used in selective classification (Geifman & El-Yaniv, 2017) to decide when to abstain from making a prediction. However, we found that both ensembling and model confidence are not well-suited for combining experts with *complementary* expertise as expert confidence on OOD, i.e., on data that they did not see during training, can be poor or misleading (Koh et al., 2020).

In this work, we propose Fusion of Experts (FoE), a simple learning problem for training a fuser using validation data from the $K$ domains that we found to perform very well in practice. Let $\{x_i^k \in \mathcal{X}, y_i^k \in \mathcal{Y}\}_{i=1}^{n_k}$ be validation data from every domain $k = 1, \ldots, K$. We construct features for every input by concatenating expert outputs, i.e., $f(x) = [f_1(x), \ldots, f_K(x)]$ and train the fuser $F_\theta$ parameterized by $\theta$ via empirical risk minimization:

$$\min_\theta \sum_k \sum_i \ell(F_\theta(f(x_i^k)), y_i^k), \tag{3.2}$$

where $\ell$ is a suitable loss function such as cross-entropy. The procedure is domain agnostic, and the fuser can be a small fully-connected neural network, that ingests the outputs of all experts to produce a class prediction.

### 3.2 Fusion of generative experts

Next, we discuss FoE for generative experts in the context of LLMs. The key difference is that the output space is a lot more complicated, e.g., natural text in LLM applications. This makes the strategies discussed above non-trivial to apply.

Ensembling LLMs can be done by averaging the next token probabilities during generation, however, this requires that all LLMs share the tokenizer and vocabulary, which is rarely true for open-source models and essentially constraints one to fine-tuning the same model for different domains to obtain experts. Confidence-based expert selection is fairly straightforward to use with LLMs by comparing the log-likelihoods (or perplexities) of LLMs generating corresponding outputs (Ren et al., 2023).

Directly applying the fusing strategy from supervised learning 3.2 would essentially mean training a new LLM, *i.e.*, a model that inputs concatenated texts generated by expert models and generates text matching the reference outputs. Jiang et al. (2023) explored such "generative fusion" in the context of a single domain and regular (non-expert) LLMs. However, training a new generative LLM fuser that would be good on all $K$ domains contradicts our problem motivation, *i.e.*, that it is challenging to train a model simultaneously suitable for all domains. Instead, we opt for a simpler fusing approach where we learn which expert to use for a given input.

First, we make a modification to how an input is represented using the expert models. Instead of concatenating their generated text outputs, we concatenate the embeddings of the input and generated tokens from the last layer of the corresponding transformer models (the predominant architecture of modern LLMs). Let $f_k^e(x)$ denote the average of $f_k$'s last-layer token embeddings of the input text $x$ and the output text $f_k(x)$. Such embeddings were previously used by Ren et al. (2023) for OOD detection with LLMs, thus suggesting that they contain information about data distributions we need to identify the correct expert.

Our representation of an input using experts is the concatenation of these averaged embeddings: $f(x) = [f_1^e(x), \ldots, f_K^e(x)]$. We train the FoE model via empirical risk minimization to predict the index of the correct expert:

$$\min_\theta \sum_k \sum_i \ell(F_\theta(f(x_i^k)), k), \tag{3.3}$$

where $\ell$ is the cross entropy loss for a $K$-way classification problem. As before, $F_\theta$ can be a simple fully-connected neural network. In this formulation, the fuser $F_\theta$ ingests a vector of concatenated embeddings $f(x)$ and outputs the index of an expert corresponding to the input $x$. The expert is then used to produce the final generation.

### 3.3 WHEN FUSING EXPERTS IS A GOOD IDEA

Using only expert outputs may seem restrictive, but it is actually not too harmful in the applications we consider. To keep things simple, we consider the task of choosing one of the experts as in 3.3. As we shall see, as long as (the expertise of) the experts are complementary, we expect their outputs to be sufficiently informative.

Consider the expert fusion problem as a multi-way hypothesis testing problem. Define $F_*(X)$ as the ground truth best expert for input $X$:

$$F_*(x) \triangleq \arg\min_{k \in [K]} \mathbf{E}\big[\ell(f_k(x), Y) \mid X = x\big] \tag{3.4}$$

and let $F(f(X))$ be an approximation to $F_*$ that only uses expert outputs as inputs. Fano's inequality provides a lower bound on the accuracy of $F \circ f$:

$$\mathbf{P}\{F(f(X)) \neq F_*(X)\} \geq \frac{H(F_*(X)) - I(f(X), F_*(X)) - \log 2}{\log(K - 1)}, \tag{3.5}$$

where $H(F_*(X))$ is the (Shannon) entropy of $F_*(X)$ and $I(f(X), F_*(X))$ is the mutual information between $f(X)$ and $F_*(X)$. From this lower bound, we see that the larger the mutual information between $F_*(X)$ and $f(X)$, the higher the accuracy we can expect from $F \circ f$. Intuitively, $I(f(X), F_*(X))$ is large whenever it is possible to recover $F_*(X)$ from expert outputs $f(X)$, and this is exactly the case when the experts are complementary.

For example, consider a classification task: the experts themselves are classifiers $f_k : \mathcal{X} \to \Delta^{C-1}$, and the label is one-hot encoded. As long as there is label shift (Lipton et al., 2018) among the domains, we expect $\mathbf{E}\big[f(X_k)\big] \approx \mathbf{E}\big[Y_k\big]$ to vary across domains. Thus it is possible to distinguish between inputs from different domains from $f(X)$ (so $I(f(X), F_*(X))$ is large), and we expect good expert selection performance.

## 4 FRUGAL FUSION OF EXPERTS (FRUGALFOE)

Producing a prediction or conditional generation output, from an input $x$ via FoE, can be summarized as follows: we first query $K$ experts, extracting a vector of concatenated outputs $f(x)$, and then use a

fuser $F_\theta$ that produces a final output based on $f(x)$. This leads to high performance when applied to test data but with the limitation that querying all experts can be costly. We now introduce a sequential algorithm called FrugalFoE to select a small yet sufficient set of experts. For the rest of this section, we assume that querying an expert $f_k$ incurs a non-negative cost $c_k$, which might include energy consumption if experts are run locally or API costs when a third party provides experts.

## 4.1 INTRODUCING FRUGALFOE

We start with some definitions and then present our approach in a general form that covers both applications to classification and generative tasks. Let $(X, Z)$ represent a random pair of inputs and outputs; at test time, the input $X$, *e.g.*, an image of a cat or a long news article, is fixed at $x$ while the output $Z$ is unknown. In this setup, $Z$ can be either a label $Y$, in the classification case, or a domain membership variable, in the generative case. Let $\mathcal{F}$ be the set of all experts $\{f_1, \cdots, f_K\}$ in the classification case and the set of all embedders obtained from experts $\{f_1^e, \cdots, f_K^e\}$ in the generative case. To ease the exposition of this section, we (i) refer to elements of $\mathcal{F}$ as "experts" in both classification and generative cases and (ii) drop the superscript "$e$" in the generative case. Furthermore, define $f_\mathcal{S}(x) \triangleq \left[f_k(x)\right]_{f_k \in \mathcal{S}}$ as the concatenation of the outputs of all experts in $\mathcal{S}$. Finally, the conditional expected loss of predicting with a set of experts $\mathcal{S}$ given the event $\{f_{\tilde{\mathcal{S}}}(X) = f_{\tilde{\mathcal{S}}}(x)\}$, *i.e.*, after querying all experts in $\tilde{\mathcal{S}}$ and observing their outputs, is defined as

$$L(x, \mathcal{S}, \tilde{\mathcal{S}}) \triangleq \mathbf{E}\Big[\ell\big(F_\theta(f_\mathcal{S}(X)), Z\big) \mid f_{\tilde{\mathcal{S}}}(X) = f_{\tilde{\mathcal{S}}}(x)\Big] + \lambda \sum_{k: f_k \in \mathcal{S}} c_k, \tag{4.1}$$

where $F_\theta$ represents a given fuser parameterized by $\theta$ and $\lambda > 0$ is a constant that controls the trade-off between querying costs and errors made by the fuser. Realize that the cost sum is taken over $k$ for $f_k \in \mathcal{S}$, *i.e.*, it depends on a set $\mathcal{S}$ of experts we are evaluating. Note that the used fuser $F_\theta$ depends on $\mathcal{S}$ through the inputs $f_\mathcal{S}(x)$. This implies that we need a fuser for each $\mathcal{S} \in 2^\mathcal{F}$. We assume that fusers for all $\mathcal{S}$ are available and discuss this aspect further in Section 4.2. Now we are ready to introduce the idea behind FrugalFoE.

Suppose that, for a fixed input $x$ and up to a certain instant, we have queried the experts $\tilde{\mathcal{S}}$. The expected loss of stopping at this point and applying a fuser on top of $f_{\tilde{\mathcal{S}}}(x)$ is given by $L(x, \tilde{\mathcal{S}}, \tilde{\mathcal{S}})$. The question is whether to query an extra expert or not: can we find an extra expert $\bar{f} \notin \tilde{\mathcal{S}}$ such that $L(x, \tilde{\mathcal{S}} \cup \{\bar{f}\}, \tilde{\mathcal{S}}) < L(x, \tilde{\mathcal{S}}, \tilde{\mathcal{S}})$? That is, *given all the information collected up to that point, can we do better by querying more experts?* If yes, we should find the extra expert $\bar{f}$ that minimizes $L(x, \tilde{\mathcal{S}} \cup \{\bar{f}\}, \tilde{\mathcal{S}})$ and query it. If not, we should stop and apply $F_\theta$ on the top of $\tilde{\mathcal{S}}$ to get a final prediction. In practice, however, we cannot evaluate the conditional expected loss $L(x, \mathcal{S}, \tilde{\mathcal{S}})$, for any sets $\mathcal{S}$ and $\tilde{\mathcal{S}}$, because the conditional distribution of $\ell\big(F_\theta(f_\mathcal{S}(X)), Y\big)$ given $\{f_\mathcal{S}(X) = f_\mathcal{S}(x)\}$ is usually unknown. In spite of that, we can estimate it on the fly using a $k$-nearest-neighbors non-parametric estimator for conditional expectations (Hastie et al., 2009) using validation data:

$$\hat{L}(x, \mathcal{S}, \tilde{\mathcal{S}}) \triangleq \frac{1}{M} \sum_{(x_m, z_m) \in \mathcal{N}_M(x, \tilde{\mathcal{S}})} \ell(F_\theta(f_\mathcal{S}(x_m)), z_m) + \lambda \sum_{k: f_k \in \mathcal{S}} c_k. \tag{4.2}$$

The neighborhood set $\mathcal{N}_M(x, \tilde{\mathcal{S}})$ are the $M$ nearest neighbors (and corresponding targets) of the input point $x$ among the validation set, where the distance between points is the Euclidean distance in the space of the queried experts' outputs, *i.e.*, $d_{\tilde{\mathcal{S}}}(x, x') = \|f_{\tilde{\mathcal{S}}}(x) - f_{\tilde{\mathcal{S}}}(x')\|_2$. Thus, we assume access to the experts' outputs for all data points in a validation set, *i.e.*, for each combination data point/expert, we have a different output.

## 4.2 IMPLEMENTING FRUGALFOE

**Starting expert.** In our problem setting we interact with input $x$ only through the expert outputs. Thus, we should select an expert to call first, which will be the same for every input. We can make this selection by simply considering the loss of 1-expert fusers on the validation data $\{x_i^k, z_i^k\}_{i=1}^{n_k}$ ($z_i^k = y_i^k$ in classification and $z_i^k = k$ in generation):

$$\arg\min_{\bar{f} \in \mathcal{F}} \sum_{i,k} \ell(F_\theta(\bar{f}(x_i^k)), z_i^k). \tag{4.3}$$

That is, we choose the expert that has the best average performance in the validation set. For interpretability reasons, we use 0-1 loss as $\ell$ in our implementation of FrugalFoE.

**Subsequent experts.** Let $\tilde{\mathcal{S}}$ be the set of experts queried so far for $x$. Before deciding on the next expert, we update the current estimate of the quality of $\tilde{\mathcal{S}}$. We use $\hat{L}(x, \tilde{\mathcal{S}}, \tilde{\mathcal{S}})$ from (4.2) as the quality estimate. Next, we find the expert $\bar{f}^*$ that we expect to provide the maximum improvement:

$$\bar{f}^* = \arg\min_{\bar{f} \in \mathcal{F} \setminus \tilde{\mathcal{S}}} \hat{L}(x, \tilde{\mathcal{S}} \cup \{\bar{f}\}, \tilde{\mathcal{S}}). \tag{4.4}$$

If $\hat{L}(x, \tilde{\mathcal{S}} \cup \{\bar{f}^*\}, \tilde{\mathcal{S}}) - \hat{L}(x, \tilde{\mathcal{S}}, \tilde{\mathcal{S}}) > 0$ we terminate the search and return $F_\theta(f_{\tilde{\mathcal{S}}}(x))$. Otherwise, we evaluate expert $\bar{f}^*$ on the input $x$, update $\tilde{S} = \tilde{S} \cup \{\bar{f}^*\}$, and continue the search. In our experiments, we consider the cost $c_k$ of all experts to equal 0.01. Then $\lambda$ can be interpreted as the minimal error rate reduction we want to achieve when deciding whether to query an additional expert.

**Obtaining fusers for subsets of experts.** So far we have assumed that we have access to fusers $F_\theta(f_{\mathcal{S}}(\cdot))$ for all $\mathcal{S} \in 2^{\mathcal{F}}$. These fusers must be trained before FrugalFoE deployment using training data and objective functions in equations 3.2 and 3.3. In general, this is only feasible for a small number of experts, or if we restrict FrugalFoE to always terminate after a small number of expert calls (out of a potentially bigger set of experts). It is, however, possible to bypass this limitation by using kNN classifiers as the fuser models as those do not require training and can be evaluated on the fly at test time. Specifically, let $x_m \in \mathcal{N}_M(x, \tilde{\mathcal{S}})$ be a point from the validation dataset that is a neighbor of a test input $x$ based on the expert outputs in $\tilde{\mathcal{S}}$. We can evaluate the loss on this point required for FrugalFoE as follows:

$$\ell(F_\theta(f_{\tilde{\mathcal{S}}}(x_m)), z_m) = \ell(\text{kNN}(x_m), z_m), \tag{4.5}$$

where $\text{kNN}(x_m)$ is the majority output corresponding to $\kappa$ nearest neighbors of $x_m$ in the validation data excluding itself. The neighbors for each validation point can be precomputed using the outputs of *all* experts $\mathcal{F}$ before deployment. Evidently, this procedure bypasses the need to train $2^{\mathcal{F}}$ fusers.

### 4.3 FRUGALFOE AS A GRAPH SHORTEST-PATH PROBLEM SOLVER

FrugalFoE is an algorithm that can be motivated as a shortest-path problem solver with connections to the $A^*$ algorithm (Russell, 2010). We first need to frame the sequential expert selection problem as a graph search problem to see that. Consider a weighted directed graph $\mathcal{G} = (\mathcal{V}, \mathcal{E})$ with $2^K + 1$ vertices: $2^K$ vertices corresponding to all subsets (including the empty set) of the $K$ experts, and an additional target vertex. We label each vertex (except the target vertex) with (the set of indices of) the subset of the experts associated with the vertex. For example, the vertex $\{1, 2\}$ is associated with the subset $\{f_1, f_2\}$. Each vertex associated with a subset of experts $\mathcal{S}$ is connected to vertices associated with the subsets of size $|\mathcal{S}| + 1$, including $\mathcal{S}$. The length of each edge between two vertices associated with subsets of the experts is the cost of querying the additional expert, scaled by $\lambda$, in the current vertex. For example, if $K = 3$, the vertex $\{1\}$ is connected to the vertices $\{1, 2\}$ and $\{1, 3\}$, and the lengths of the edges are $\lambda c_2$ and $\lambda c_3$. Finally, all $2^K$ vertices associated with subsets of the experts are connected to the terminal vertex. For a given input $x$ and set of fusers, the length of the edge connecting a vertex associated with a set of experts $\mathcal{S}$ to the terminal vertex is $\frac{1}{M} \sum_{(x_m, z_m) \in \mathcal{N}_M(x, \tilde{\mathcal{S}})} \ell(F_\theta(f_{\mathcal{S}}(x_m)), z_m)$ if we have gathered outputs from all experts in $\tilde{\mathcal{S}}$ (4.2).

From here, we can draw connections to the $A^*$ algorithm. In $A^*$, for any prospective vertex $n$ to be visited on the graph, the estimated distance traversing through $n$ from the initial to the terminal vertex can be dissected into the sum of the known distance from the initial vertex to $n$, denoted as $g(n)$, and an estimate of the distance from $n$ to the terminal vertex, denoted as $h(n)$. Among all prospective vertices, we opt for $n^*$ that yields the smallest sum $g(n^*) + h(n^*)$. In FrugalFoE, if we are currently at a vertex representing the set of experts $\tilde{\mathcal{S}}$, the next step is to query some extra expert $\bar{f}$ or opt for the terminal vertex, which we denote by $T$. If we decide to query $\bar{f}^*$, using the logic explained in equation 4.4, then $g(\tilde{\mathcal{S}}, \bar{f}^*) = \lambda \sum_{k: f_k \in \tilde{\mathcal{S}} \cup \{\bar{f}^*\}} c_k$ and $h(\tilde{\mathcal{S}}, \bar{f}^*) = \frac{1}{M} \sum_{(x_m, z_m) \in \mathcal{N}_M(x, \tilde{\mathcal{S}})} \ell(F_\theta(f_{\mathcal{S}}(x_m)), z_m)$. If we decide on the terminal vertex (i.e., to conclude the search), $g(\tilde{\mathcal{S}}, T) = \hat{L}(x, \tilde{\mathcal{S}}, \tilde{\mathcal{S}})$ and $h(\tilde{\mathcal{S}}, T) = 0$. The connection between FrugalFoE and the $A^*$ algorithm is clear once we define $g$ and $h$; however, there is no perfect correspondence between the two algorithms since in FrugalFoE the functions $g$ and $h$ depend on the current set of vertices $\tilde{\mathcal{S}}$ as opposed to the classical $A^*$ version where these functions only depend on the candidate vertex. This difference is mainly due to the fact that we can not "undo" querying an expert, so it is more reasonable to try to utilize all expert outputs available at any given decision point.

## 5   EXPERIMENTS

We evaluated the efficacy of the FoE method using a wide range of experiments, including image classification, summarization, text generation evaluation, sentiment analysis, and massive multitask language understanding (MMLU) (Hendrycks et al., 2021). Additionally, we investigated how to enhance FoE's efficiency, aiming to achieve the desired accuracy while consulting the fewest number of experts, i.e., FrugalFoE. FoE manages to achieve close (and matches on some tasks) performance compared to the "*oracle model*" (*i.e.*, always selecting the most suitable expert for a given task) and consistently surpasses individual experts and other popular baseline methods (including FrugalML (Chen et al., 2020), ensemble) by a notable margin. Interestingly, FoE inherently tends to be frugal in summarization and sentiment analysis tasks. Specifically, relying on information from just one expert often yields results as effective as consulting all experts. For tasks where FoE is not naturally frugal, our FrugalFoE algorithm proves effective. For instance, in the CIFAR experiment, FrugalFoE queries only 37.5% of the experts to match the accuracy achieved by querying all experts.

### 5.1   THE CIFAR-100 SUPER-CLASS CLASSIFICATION TASK

We start with an image classification experiment on CIFAR-100. While this is a relatively simple task and on a small scale, it effectively demonstrates the capabilities of FoE.

We partition CIFAR-100 into 20 sections where each section contains all images from 30 sub-classes (Krizhevsky et al., 2009). Out of these 30 sub-classes, 20 come exclusively from one sub-class of each super-class (for instance, *beaver* from *aquatic mammals*, *bottles* from *food containers*, etc), while the remaining 10 are picked at random from all the other sub-classes. Thus, these sections have overlapping images, meaning they are not mutually exclusive. This partitioning strategy (inspired partly by Santurkar et al. (2020)) ensures that each expert excels in classifying one sub-class within each super-class, however, they have some shared knowledge (more details in Appendix A.2).

We train 20 experts, each of them using ResNet-18 on their own data partition to predict super-classes (He et al., 2016). We use $40k$ images from the CIFAR-100 training set for partitioning and expert training and hold $10k$ images from the training set out as a validation set to train our fusing strategy by solving equation 3.2. The final test accuracy is measured using the CIFAR test set.

Our results are shown in Table 1. In the confidence-based fusing baseline, we used the maximum softmax score as an indicator of a model's confidence and then chose the model with the highest confidence for predictions, where models were trained with mix-up, a method that effectively calibrates the model's confidence (Guo et al., 2017; Zhang et al., 2018). Additionally, we show the ac-

Table 1: CIFAR-100 super-class classification.

| Method | Final Acc. (%) | Expert Selection Acc. (%) |
|---|---|---|
| FoE | **82.13** | **75.27** |
| Experts Average | $50.04_{\pm 1.56}$ | – |
| Confidence-based Fusion | 74.07 | 59.69 |
| Ensemble | 76.64 | – |
| Oracle Expert | 87.63 | 100 |

curacy of an oracle classifier, which operates under the assumption that each test sample is always evaluated using the ideal expert — that is, the expert trained on the data containing the subclass of the the test sample. Our results demonstrate that using FoE markedly improves accuracy over individual experts. FoE also outperforms the confidence-based fusion approach and a basic ensemble method, both in prediction accuracy and expert selection. Notably, FoE's performance comes close to matching the accuracy of the oracle expert.

**FrugalFoE for image classification.** FoE performs well on the CIFAR task, however, for each test sample, it needs to query all 20 experts, which is computationally expensive. We seek to make it frugal using our FrugalFoE algorithm described in Section 4.2 using the kNN fusing approach 4.5 with $\kappa = \{7, 9, 11\}$ (FoE's performance does not seem to be sensitive to different $\kappa$ values). We vary the selection of $M$ and $\lambda$ to control the number of experts we end up with querying. The FrugalMoE results are shown in Figure 2 where one can observe that FrugalFoE can use 37.5% of the experts to reach the accuracy of using the entire 20 experts on the CIFAR task.

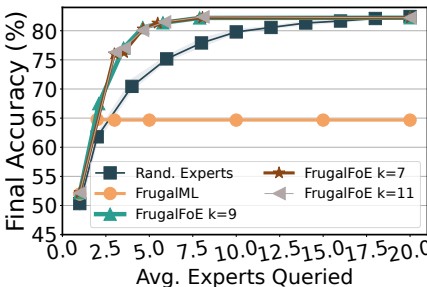

Figure 2: Frugal CIFAR-100 w/ various $\kappa$.

When querying the same number of experts, FrugalFoE outperforms FrugalML, which is also limited to up to two experts. In Figure 5 we present results with a neural network as a fuser where we limit the maximum number of expert calls to 5 to make the problem feasible.

## 5.2 THE SENTIMENT ANALYSIS TASK

We consider language model applications, starting with the sentiment analysis task. We use fine-tuned sentiment classification models from Hugging Face (more details in Appendix A.4) as experts and the corresponding datasets, Twitter Sentiment Analysis, Twitter Financial News, Poem Sentiment, Data Reviews Sentiment Anal-

Table 2: Sentiment analysis task experiments.

| Method | TFN | Poem | Auditor | Reviews Sent. | Avg. |
|---|---|---|---|---|---|
| FoE | 87.54% | 85.71% | 81.71% | 95.20% | **91.88**% |
| TFN Expert | 87.54% | 0.0% | 58.54% | 0.40% | 26.85% |
| Poem Expert | 6.69% | 85.71% | 15.24% | 59.76% | 46.95% |
| Auditor Expert | 51.98% | 45.71% | 81.71% | 49.65% | 53.50% |
| Reviews Sent. Expert | 71.12% | 62.86% | 32.93% | 95.20% | 79.44% |
| FoE w/ TFN Expert features only | 86.93% | 85.71% | 82.32% | 95.20% | 91.81% |
| FoE w/ Poem Expert features only | 87.23% | 82.86% | 81.10% | 95.20% | 91.75% |

ysis, Auditor Sentiment (Naji, 2012; Sheng & Uthus, 2020). Though sentiment analysis is essentially a classification task, we train the fuser using the generative model strategy 3.3. To test we combined the test sets from all tasks. Results for per task and average accuracies are in Table 2 (upper part). FoE achieves 99.1% accuracy in expert selection. FoE almost reaches the best sentiment classification accuracy on each downstream task, *i.e.*, the oracle expert performance.

Our other finding is that the expert outputs, i.e., the embeddings of language models are highly informative such that only querying a single expert is sufficient for the FoE to predict the expert-to-use surprisingly well. In Table 2 (lower part) we present results when using a single expert, i.e., extreme frugal case, noting that performance is almost equal to using all experts. Results using other stand-alone experts are in Table 13.

## 5.3 THE SUMMARIZATION TASK

In this section, we present our experiments on generative tasks, starting with the summarization task. We use fine-tuned Pegasus models on six downstream summarization tasks as experts (Zhang et al., 2020). We use the combined validation datasets from all six summarization tasks to train the generative fuser 3.3 for expert prediction. For testing, we combined the test sets from all tasks and used ROUGE-2 (Lin & Hovy, 2003) scores to assess the performance. Results for per task and average accuracies are in Table 3 (upper part). Expert selection accuracy of FoE is 99.1% with the six experts. One can see that FoE almost reaches the best ROUGE-2 score on each downstream task, *i.e.*, FoE almost reaches the performance of the oracle expert.

Table 3: Summarization task experiments, ROUGE-2 score (↑) as the evaluation metric.

| Method | CNN DM | XSUM | Multi-News | BillSum | Big-Patent | AESLC | Avg. |
|---|---|---|---|---|---|---|---|
| FoE | 20.2225 | 23.8662 | 18.3499 | 36.9398 | 27.0756 | 20.6736 | **23.7171** |
| CNN DM Expert | 20.2472 | 3.5595 | 8.2049 | 14.0173 | 12.7726 | 2.4269 | 11.3757 |
| XSUM Expert | 7.4224 | 23.8904 | 4.0622 | 8.0220 | 11.0968 | 2.9332 | 11.9509 |
| Multi-News Expert | 10.3799 | 4.8769 | 18.5120 | 7.2528 | 7.0045 | 0.4224 | 8.5559 |
| BillSum Expert | 8.7146 | 3.2458 | 4.6663 | 38.5739 | 6.2896 | 1.9565 | 8.1622 |
| Big-Patent Expert | 7.2759 | 3.4595 | 4.1817 | 10.4013 | 27.0760 | 0.9100 | 10.2426 |
| AESLC Expert | 5.9838 | 3.6185 | 1.6123 | 4.1015 | 3.0258 | 20.6842 | 4.6352 |
| FoE w/ CNN DM Expert features only | 20.1599 | 23.7911 | 18.1450 | 37.8557 | 27.0765 | 20.5473 | 23.7180 |
| FoE w/ XSUM Expert features only | 20.1402 | 23.8484 | 18.0158 | 36.3483 | 27.0709 | 20.5520 | 23.5963 |
| FoE w/ Multi-News Expert features only | 19.8900 | 23.3156 | 17.7478 | 18.1870 | 27.0660 | 20.3217 | 21.9752 |

Similar to our sentiment analysis experiment, querying a single expert is almost as good as using all experts (see Table 3 lower part and Table 14 for the remaining experts).

## 5.4 THE MMLU TASK

In all previous experiments, we were fusing experts properly fine-tuned for their respective domains. Here we consider a weaker notion of experts, i.e., general LLMs that happen to perform better than others on a particular domain. Specifically, we consider the MMLU (Hendrycks et al., 2021) multiple-choice QA task which consists of 57 categories (elementary mathematics, US history, etc.). As experts, we use 15 open-source LLMs with sizes of ∼ 7 billion parameters from the Open LLM Leaderboard (Beeching et al., 2023). We consider an LLM with the highest accuracy on a category to be an expert for this category. FoE is trained as a generative fuser 3.3 (details are in Appendix A.3). The test set is the combination of test sets from all 57 categories.

The embedding dimension of expert LLMs is fairly high (4096), while there is only a total of 1700 samples for training the fuser, thus we use single expert outputs for FoE in this experiment. Results are presented in Table 4. We see that the results with the weak experts are worse as we no longer can match the oracle performance, however, FoE still outperforms the strongest of the considered LLMs. Next, unlike our previous experiments, we notice a discrepancy in performance when using outputs of different

Table 4: MMLU with weak experts.

| Method | Expe. Selec. Acc. | Overall |
|---|---|---|
| FoE (w/ Expert 1 features only) | **74.8%** | **49.85** |
| FoE (w/ Expert 2 features only) | 45.4% | 48.27 |
| FoE (w/ Expert 3 features only) | 51.9% | 48.54 |
| Avg. Experts | – | $41.34_{\pm 7.22}$ |
| The Best Expert | – | 47.35 |
| Oracle Expert | 100% | 51.56 |

weak experts, especially in terms of expert selection accuracy. Overall, we conclude that weak experts are not as effective, although still can benefit from FoE. We also tested the generalization capability of FoE to unseen domains on the MMLU task, the results can be found in Appendix B.1.

## 5.5 THE TEXT GENERATION EVALUATION TASK

We investigate the potential of using complementary experts to evaluate machine-generated summaries. For this experiment, we use human-annotated summary datasets namely SummEval (Fabbri et al., 2021), Newsroom (Grusky et al., 2018), QAGS-CNN/XSUM (Wang

Table 5: Correlation of human labels and automated evaluation metrics.

| Method | SummEval | NEWSROOM | H-XSUM | Q-CNN | Q-XSUM | Avg. |
|---|---|---|---|---|---|---|
| FoE | 0.613 | 0.652 | 0.237 | 0.738 | 0.343 | **0.517** |
| SummEval Expert | 0.655 | 0.51 | 0.209 | 0.716 | 0.357 | 0.489 |
| NEWSROOM Expert | 0.293 | 0.685 | 0.167 | 0.673 | 0.112 | 0.386 |
| H-XSUM Expert | 0.548 | 0.592 | 0.241 | 0.747 | 0.348 | 0.495 |
| Q-CNN Expert | 0.502 | 0.612 | 0.237 | 0.747 | 0.337 | 0.487 |
| Q-XSUM Expert | 0.499 | 0.586 | 0.243 | 0.729 | 0.323 | 0.476 |

et al., 2020a), and HALL-XSUM (Maynez et al., 2020). Each dataset has human ratings for specific dimensions such as coherence and consistency. To train and evaluate experts for each one of the datasets/domains, we: (i) extract some automatic metrics, *e.g.*, BARTScore (Yuan et al., 2021), BERTScoreFree (Shen et al., 2022), UniEval (Zhong et al., 2022), from summaries; (ii) randomly select training and test points from each dataset; (iii) for each pair domain/dimension, we train a linear model (expert) that optimally combines precomputed metrics. We compare the performances of FoE and individual experts in evaluating the consistency[1] (or factuality) of machine-generated summaries. In Table 5, we can see that, for individual tasks, FoE does not lead to the highest correlations; however on the aggregated test set (last column) our approach delivers the overall best results. We present additional details and comparison to individual metrics in Appendix A.1.

We note that our FrugalFoE approach is not directly applicable in this setting due to the design of the experts. In this case, they are simple linear models that use various automatic metrics as features. The experts themselves are cheap to evaluate, however, it may be desirable to reduce the number of their input features, i.e., automatic metrics based on LLMs, to improve test-time efficiency. We leave the extension of FrugalFoE to such expert design for future work.

## 6 CONCLUSION

Our work demonstrates the benefits of fusing experts fine-tuned for different data subdomains. Obtaining high-quality experts has become relatively easy with the emergence of foundation models, thanks to their few-shot learning capabilities. The fusing procedure is also fairly simple: using appropriate model outputs it suffices to train a simple neural network, as supported by our experiments. We have seen that it is even beneficial to fuse general LLMs that happen to perform well on various domains, although not as effective as the fusion of proper expert models.

We have also proposed a frugal extension of the expert fusion for the cases where querying all experts at test time is too costly. Our solution based on kNN classifiers and the graph shortest path analogy allowed us to bypass the combinatorial nature of the problem that limited prior frugal methods to consider at most 2-3 expert/API calls (Chen et al., 2020; 2023b). Interestingly, in the considered LLM applications, using only the outputs of an arbitrary single expert for all test inputs was sufficient to identify the correct expert for the input. This observation suggests that LLM embeddings contain a lot of information that can potentially be utilized in other applications. This observation extends the findings of Ren et al. (2023), who demonstrated that LLM embeddings are good for OOD detection, i.e., distinguishing their own domain from everything else.

---

[1]Consistency/factuality measure if the facts presented in the summary are consistent with those in the source text. In the Newsroom dataset, this dimension is termed as "relevance". See Table 6 for the full set of results.

## ACKNOWLEDGMENTS

We thank the anonymous reviewers for their valuable insights and recommendations, which have greatly improved our work. This research has been graciously funded by NGA HM04762010002, NSF IIS1955532, NSF CNS2008248, NIGMS R01GM140467, NSF IIS2123952, NSF DMS2027737, NSF BCS2040381, NSF DMS2112273, NSF IIS2311990, Semiconductor Research Corporation (SRC) AIHW award 2024AH3210, and DARPA ECOLE HR00112390063.

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

CONTENTS OF THE APPENDIX

## A    MORE DETAILS ON EXPERIMENTAL SETUPS

### A.1    MORE DETAILS ON TEXT GENERATION EVALUATION

The first observation we need to make is that, because each dataset has a different format for the human annotations, experts' outputs are not comparable, and we cannot compute the aggregate correlation of predictions and human labels for a given task (pair dataset/dimension) when evaluating FoE. Instead, we estimate the expected conditional correlation between predictions and human labels in the following way: using the outputs from the eleven experts and for each domain, we classify each text in the test set according to their dataset membership, and according to the empirical distribution of classes induced by the classifier, we compute a weighted average of the correlations achieved by experts in that specific task. Now, we can describe our experiment in detailed steps.

This experiment is repeated 25 times with different random seeds. The final correlations/numbers are the averages across the 25 repetitions. For each random seed $b$, we do the following:

1. For all datasets, extract automatic metrics, *i.e.*, BARTScore (Yuan et al., 2021), BERTScore.Free (Zhang et al., 2019b; Shen et al., 2022), Unieval (Zhong et al., 2022). For BARTScore, we use four variations: the first one is BARTScore-CNN (used by Yuan et al. (2021)), and for the last three, we use Pegasus (Zhang et al., 2020) pre-trained on CNN-DM, Newsroom, and XSUM as the backbone model. We work only with reference-free metrics, and therefore we do not use UniEval to evaluate relevance;

2. For each dataset, randomly select training and test samples. We select 50 test points for all datasets, but the number of training points depends on how big the datasets are. The biggest training set has 370 data points;

3. For each task (pair dataset/dimension), train a linear model using non-negative least squares to predict human rating from automatic metrics. In total, we have eleven experts;

4. Evaluate the performance (Pearson correlation) of experts and individual metrics on each one of the tasks using the test sets;

5. Append all training sets with the eleven experts' outputs as columns and train a CatBoost classifier (Prokhorenkova et al., 2018) to predict from each dataset each point has come. Because the test set is balanced, we weigh the loss to guarantee that each class has the same importance. The average confusion matrix (across Monte Carlo repetitions) can be seen in Figure 3;

6. To evaluate FoE we do the following for each task (pair dataset/dimension): using the outputs from the eleven experts and for each domain, we classify each text in the test set using the trained CatBoost classifier according to their dataset membership; according to the empirical distribution of classes induced by the classifier, we compute a weighted average of the correlations achieved by experts in that specific task. For example, if task="summeval_coherence" and the classifier outputs the class "summeval" $90\%$ of the times and the class "newsroom" $10\%$ of the times, the FoE score on that task will be $.9 \times \rho_{SE} + .1 \times \rho_{NR}$, where $\rho_{SE}$ is the SummEval's expert score and $\rho_{NR}$ is the Newsroom's expert score in that specific task;

In Table 6, we can see the full set of results. The row "avg" computes the average of all available methods, while "avg_con" only takes into account the dimension "consistency" (factuality), and

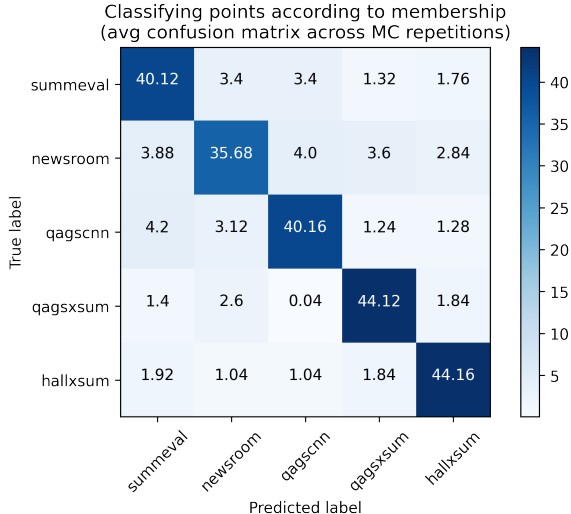

Figure 3: Average confusion matrix across Monte Carlo repetitions.

"avg_no_con" takes into account all dimensions except "consistency". We show consistency results in the main text because it is a dimension present in all datasets. For the "oracle" columns, we always select the correct expert, instead of running the classifier.

Table 6: Summarization evaluation (Pearson correlation of labels and predictions). In this table, "coh" stands for coherence, "con" stands for consistency, "flu" stands for fluency, and "rel" stands for relevance. For an explanation of these dimensions, see Yuan et al. (2021); Zhong et al. (2022).

| Tasks / Methods | oracle | FoE | summeval | newsroom | qagscnn | hallxsum | qagsxsum | unieval | bert | bart_cnndm | peg_cnndm | peg_news | peg_xsum |
|---|---|---|---|---|---|---|---|---|---|---|---|---|---|
| summeval_coh | 0.61 | 0.599 | 0.61 | 0.525 | - | - | - | 0.423 | 0.448 | 0.428 | 0.463 | 0.371 | 0.489 |
| newsroom_coh | 0.653 | 0.649 | 0.627 | 0.653 | - | - | - | 0.604 | 0.643 | 0.621 | 0.543 | 0.601 | 0.435 |
| summeval_con | 0.655 | 0.613 | 0.655 | 0.293 | 0.502 | 0.548 | 0.499 | 0.436 | 0.237 | 0.314 | 0.356 | 0.189 | 0.646 |
| newsroom_con | 0.685 | 0.652 | 0.51 | 0.685 | 0.612 | 0.592 | 0.586 | 0.615 | 0.657 | 0.634 | 0.529 | 0.589 | 0.41 |
| hallxsum_con | 0.243 | 0.237 | 0.209 | 0.167 | 0.237 | 0.241 | 0.243 | 0.191 | 0.202 | 0.203 | 0.202 | 0.216 | 0.182 |
| qagscnn_con | 0.747 | 0.738 | 0.716 | 0.673 | 0.747 | 0.747 | 0.729 | 0.627 | 0.728 | 0.737 | 0.724 | 0.598 | 0.642 |
| qagsxsum_con | 0.348 | 0.343 | 0.357 | 0.112 | 0.337 | 0.348 | 0.323 | 0.1 | 0.232 | 0.196 | 0.196 | 0.086 | 0.34 |
| summeval_flu | 0.594 | 0.578 | 0.594 | 0.459 | - | - | - | 0.339 | 0.225 | 0.281 | 0.303 | 0.155 | 0.461 |
| newsroom_flu | 0.61 | 0.599 | 0.525 | 0.61 | - | - | - | 0.553 | 0.597 | 0.579 | 0.51 | 0.565 | 0.392 |
| summeval_rel | 0.565 | 0.557 | 0.565 | 0.494 | - | - | - | 0.458 | 0.398 | 0.427 | 0.441 | 0.331 | 0.415 |
| newsroom_rel | 0.743 | 0.733 | 0.665 | 0.743 | - | - | - | 0.669 | 0.663 | 0.629 | 0.583 | 0.626 | 0.166 |
| avg | 0.587 | **0.573** | 0.549 | 0.492 | 0.487 | 0.495 | 0.476 | 0.456 | 0.457 | 0.459 | 0.441 | 0.393 | 0.416 |
| avg_con | 0.536 | **0.517** | 0.489 | 0.386 | 0.487 | 0.495 | 0.476 | 0.394 | 0.411 | 0.417 | 0.401 | 0.336 | 0.444 |
| avg_no_con | 0.629 | **0.619** | 0.598 | 0.581 | - | - | - | 0.508 | 0.496 | 0.494 | 0.474 | 0.442 | 0.393 |

## A.2 More details on the CIFAR experiment

In this section, we provide more information about the CIFAR-100 experiment presented in the main paper. To construct features for training the fusing strategy in FoE (*i.e.*, a classifier to directly predict labels), we use the softmax scores of the last layer for each expert. The detailed partitions used in our CIFAR-100 experiments are shown in Table 7. The overlapping of sub-classes among partitions is shown in Figure 4.

Table 7: Partition details of the CIFAR-100 experiment.

| Expert | Sub-class Contained |
|---|---|
| Expert 1 | {30, 73, 62, 9, 0, 87, 25, 7, 3, 12, 23, 19, 66, 99, 35, 27, 50, 96, 8, 81, 2, 63, 10, 69, 41, 79, 97, 5, 28, 22} |
| Expert 2 | {95, 1, 70, 28, 0, 87, 94, 7, 42, 12, 71, 38, 75, 77, 35, 93, 50, 96, 13, 89, 14, 25, 17, 18, 59, 19, 73, 88, 45, 41} |
| Expert 3 | {55, 67, 62, 16, 0, 22, 84, 24, 43, 12, 49, 21, 34, 45, 2, 29, 50, 96, 8, 41, 81, 98, 78, 79, 56, 38, 14, 85, 59, 76} |
| Expert 4 | {72, 1, 92, 28, 83, 87, 84, 6, 3, 12, 33, 15, 75, 26, 11, 29, 65, 56, 58, 85, 99, 16, 8, 97, 47, 0, 54, 81, 2, 96} |
| Expert 5 | {55, 1, 70, 10, 51, 86, 94, 24, 97, 12, 33, 19, 64, 77, 35, 44, 50, 59, 58, 81, 4, 73, 45, 40, 79, 61, 83, 46, 7, 88} |
| Expert 6 | {55, 32, 92, 9, 0, 22, 25, 24, 88, 12, 23, 21, 34, 77, 2, 29, 65, 56, 48, 69, 50, 96, 49, 52, 40, 93, 70, 44, 58, 83} |
| Expert 7 | {72, 67, 92, 9, 57, 86, 25, 24, 43, 37, 23, 19, 34, 26, 35, 78, 74, 47, 90, 69, 71, 98, 66, 30, 75, 79, 64, 85, 58, 52} |
| Expert 8 | {95, 1, 62, 16, 83, 39, 84, 6, 88, 12, 33, 31, 64, 79, 46, 27, 36, 59, 58, 89, 81, 53, 4, 17, 71, 5, 65, 72, 14, 48} |
| Expert 9 | {95, 1, 92, 28, 51, 22, 94, 14, 42, 17, 49, 19, 75, 99, 46, 78, 50, 59, 58, 41, 84, 69, 40, 93, 73, 18, 85, 83, 33, 23} |
| Expert 10 | {4, 67, 62, 61, 57, 40, 20, 18, 97, 76, 23, 38, 63, 77, 98, 29, 74, 56, 58, 41, 66, 71, 90, 21, 53, 78, 45, 6, 11, 89} |
| Expert 11 | {30, 1, 62, 9, 83, 87, 94, 18, 88, 37, 71, 31, 66, 99, 2, 93, 80, 52, 48, 69, 50, 64, 59, 81, 23, 55, 41, 10, 17, 90} |
| Expert 12 | {4, 67, 54, 28, 0, 87, 25, 6, 88, 17, 33, 38, 66, 26, 11, 29, 74, 47, 13, 69, 49, 62, 5, 7, 80, 14, 84, 56, 32, 96} |
| Expert 13 | {55, 67, 62, 28, 0, 22, 84, 7, 88, 76, 71, 15, 75, 99, 2, 29, 74, 52, 8, 81, 72, 45, 14, 34, 24, 21, 47, 1, 82, 83} |
| Expert 14 | {30, 32, 54, 28, 53, 39, 20, 18, 43, 17, 71, 21, 64, 99, 11, 78, 65, 47, 58, 41, 73, 80, 95, 55, 2, 9, 25, 46, 15, 57} |
| Expert 15 | {30, 67, 54, 16, 57, 86, 25, 7, 97, 17, 33, 19, 64, 99, 35, 78, 36, 59, 48, 41, 94, 80, 73, 40, 50, 72, 23, 5, 14, 62} |
| Expert 16 | {95, 32, 54, 9, 57, 87, 84, 14, 43, 76, 23, 15, 34, 79, 2, 93, 50, 56, 90, 41, 72, 62, 33, 91, 99, 8, 55, 16, 22, 47} |
| Expert 17 | {72, 32, 70, 9, 51, 86, 84, 18, 42, 76, 60, 19, 64, 77, 46, 29, 65, 56, 58, 41, 5, 79, 6, 62, 34, 50, 54, 3, 99, 16} |
| Expert 18 | {30, 67, 54, 61, 53, 86, 94, 24, 43, 12, 60, 15, 64, 79, 35, 44, 80, 47, 13, 81, 69, 20, 66, 45, 10, 5, 59, 85, 83, 36} |
| Expert 19 | {4, 91, 82, 9, 53, 87, 5, 14, 3, 37, 33, 38, 66, 26, 35, 78, 80, 52, 90, 85, 32, 83, 81, 43, 97, 92, 22, 67, 21, 72} |
| Expert 20 | {4, 1, 70, 61, 83, 87, 20, 14, 88, 17, 49, 31, 75, 77, 2, 93, 74, 47, 90, 85, 35, 69, 94, 36, 86, 76, 56, 84, 59, 91} |

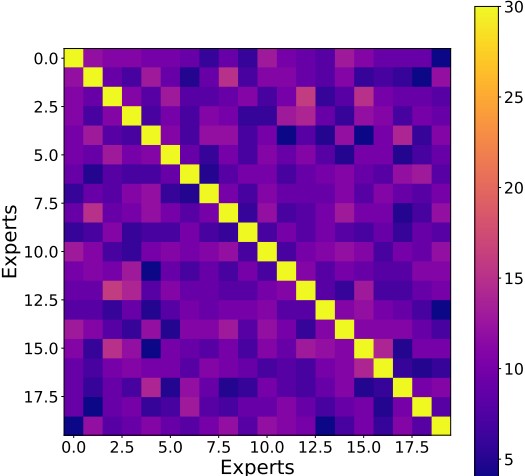

Figure 4: Overlapping among sub-classes among the 20 partitions/experts.

## A.3 More details on the MMLU experiment

In this section, we provide more information about the MMLU experiment presented in the main paper. To construct the feature to learn the fusing strategy, we use the input (without the answer choices) in the MMLU test as the prompt and generate output with maximum sequence length of 16.

We average across the sequence length dimension (prompt and generated tokens) of the last decoder block and use that as the feature for each expert.

**The selected LLMs.** The selected LLMs in our experiments are:

- `Aspik101/trurl-2-7b-pl-instruct_unload`
- `Charlie911/vicuna-7b-v1.5-lora-mctaco`
- `Fredithefish/RedPajama-INCITE-Chat-3B-Instruction-Tuning-with-GPT-4`
- `GOAT-AI/GOAT-7B-Community`
- `TheTravellingEngineer/bloom-1b1-RLHF`
- `ashercn97/manatee-7b`
- `garage-bAInd/Platypus2-7B`
- `golaxy/gogpt-7b-bloom`
- `julianweng/Llama-2-7b-chat-orcah`
- `lmsys/vicuna-7b-v1.3`
- `lmsys/vicuna-7b-v1.5-16k`
- `medalpaca/medalpaca-7b`
- `rombodawg/LosslessMegaCoder-llama2-7b-mini`
- `togethercomputer/GPT-JT-6B-v0`
- `togethercomputer/GPT-JT-6B-v1`

**The complete category scores.** The complete MMLU category scores for average experts, FoE, and the oracle expert are reported in Table 8 and Table 9.

Table 8: Detailed accuracy of all the MMLU categories (Part-1).

| MMLU Category | Average Experts | Oracle Expert | FoE |
|---|---|---|---|
| abstract algebra | 26.46 | 41.41 | 38.38 |
| anatomy | 44.58 | 65.67 | 60.45 |
| astronomy | 44.02 | 62.91 | 58.28 |
| business ethics | 42.96 | 51.52 | 45.45 |
| clinical knowledge | 48.84 | 57.58 | 59.47 |
| college biology | 46.06 | 58.04 | 54.55 |
| college chemistry | 34.75 | 45.45 | 36.36 |
| college computer science | 36.77 | 45.45 | 40.40 |
| college mathematics | 33.40 | 45.45 | 42.42 |
| college medicine | 41.43 | 53.49 | 50.58 |
| college physics | 26.47 | 44.55 | 30.69 |
| computer security | 52.59 | 69.70 | 67.68 |
| conceptual physics | 39.09 | 44.87 | 36.75 |
| econometrics | 30.80 | 40.71 | 36.28 |
| electrical engineering | 42.96 | 58.33 | 53.47 |
| elementary mathematics | 29.53 | 32.89 | 32.89 |
| formal logic | 11.25 | 16.0 | 15.2 |
| global facts | 34.07 | 44.44 | 39.39 |
| high school biology | 48.95 | 57.93 | 55.66 |
| high school chemistry | 35.54 | 41.09 | 44.06 |
| high school computer science | 42.36 | 54.55 | 56.57 |
| high school european history | 54.11 | 68.90 | 68.29 |
| high school geography | 55.23 | 65.99 | 65.48 |
| high school government and politics | 62.05 | 78.13 | 77.08 |

Table 9: Detailed accuracy of all the MMLU categories (Part-2).

| MMLU Category | Average Experts | Oracle Expert | FoE |
|---|---|---|---|
| high school macroeconomics | 42.28 | 49.36 | 48.33 |
| high school mathematics | 26.02 | 30.48 | 26.77 |
| high school microeconomics | 42.03 | 49.79 | 51.90 |
| high school physics | 31.56 | 40.0 | 36.0 |
| high school psychology | 59.22 | 70.40 | 67.65 |
| high school statistics | 37.12 | 47.44 | 42.79 |
| high school us history | 54.98 | 70.94 | 67.49 |
| high school world history | 57.37 | 75.42 | 74.58 |
| human aging | 49.31 | 61.71 | 56.76 |
| human sexuality | 50.36 | 63.08 | 57.69 |
| international law | 55.39 | 71.67 | 69.17 |
| jurisprudence | 50.65 | 67.29 | 60.75 |
| logical fallacies | 47.65 | 59.26 | 59.26 |
| machine learning | 32.31 | 44.14 | 40.54 |
| management | 59.15 | 73.53 | 71.57 |
| marketing | 30.50 | 37.77 | 36.91 |
| medical genetics | 52.05 | 69.70 | 68.69 |
| miscellaneous | 56.50 | 69.69 | 68.29 |
| moral disputes | 0.29 | 0.58 | 0.58 |
| moral scenarios | 25.41 | 30.65 | 30.65 |
| nutrition | 48.02 | 58.69 | 57.38 |
| philosophy | 48.71 | 60.97 | 60.97 |
| prehistory | 47.95 | 59.13 | 59.13 |
| professional accounting | 34.45 | 40.21 | 39.50 |
| professional law | 34.22 | 43.12 | 43.12 |
| professional medicine | 44.85 | 59.78 | 59.41 |
| professional psychology | 42.54 | 52.21 | 49.92 |
| public relations | 50.28 | 65.14 | 60.55 |
| security studies | 50.52 | 66.80 | 62.70 |
| sociology | 57.47 | 70.5 | 64.5 |
| us foreign policy | 61.01 | 74.75 | 74.75 |
| virology | 38.51 | 49.70 | 48.48 |
| world religions | 26.59 | 33.53 | 33.53 |

**MMLU scores on all individual LLMs experimented in our experiments.** The MMLU scores of individual experts used in our experiments are reported in Table 10.

A.4   MORE DETAILS ON THE SENTIMENT ANALYSIS EXPERIMENT

We select the models with their associated datasets available on Hugging Face for the sentiment analysis experiments. To train the fusing strategy, we use the input embedding (averaged across the context length dimension) as the feature. Similar to the summarization experiment, the true label indicates the source downstream task of the input text sequence/article.

**The selected models.**

- nickmuchi/finbert-tone-finetuned-fintwitter-classification

- joheras/clasificador-poem-sentiment

- FinanceInc/auditor_sentiment_finetuned

- Kaludi/Reviews-Sentiment-Analysis

Table 10: MMLU scores of our experimented individual experts.

| MMLU Category | Overall MMLU Score |
|---|---|
| Aspik101/trurl-2-7b-pl-instruct_unload | 45.9993 |
| Charlie911/vicuna-7b-v1.5-lora-mctaco | 45.8491 |
| Fredithefish/RedPajama-INCITE-Chat-3B-Instruction-Tuning-with-GPT-4 | 25.8348 |
| GOAT-AI/GOAT-7B-Community | 46.3568 |
| TheTravellingEngineer/bloom-1b1-RLHF | 25.3128 |
| ashercn97/manatee-7b | 45.9349 |
| garage-bAInd/Platypus2-7B | 47.3507 |
| golaxy/gogpt-7b-bloom | 31.9056 |
| julianweng/Llama-2-7b-chat-orcah | 44.6764 |
| lmsys/vicuna-7b-v1.3 | 44.6121 |
| lmsys/vicuna-7b-v1.5-16k | 45.8062 |
| medalpaca/medalpaca-7b | 39.4995 |
| rombodawg/LosslessMegaCoder-llama2-7b-mini | 46.2496 |
| togethercomputer/GPT-JT-6B-v0 | 42.8531 |
| togethercomputer/GPT-JT-6B-v1 | 41.9020 |

**The selected sentiment analysis dataset.** For the Twitter Financial News Sentiment and Auditor Sentiment datasets, there is no split between validation and test sets. We thus split $60\%$ of the data as the validation set and $40\%$ of the data as the test set.

- zeroshot/twitter-financial-news-sentiment
- poem_sentiment
- Kaludi/data-reviews-sentiment-analysis
- FinanceInc/auditor_sentiment

**Details on the semantic label alignment.** The goal of the semantic label alignment is to make sure all experimented models are aligned on the predicted labels. Our objective is to make the "0" for "negative" (or equivalent) and "1" for "positive" (or equivalent) for the adjusted and aligned labels. See Table 11.

Table 11: Label alignment in the sentiment prediction experiments

| Expert | Original Labels | Aligned Labels |
|---|---|---|
| Twitter Financial News Sentiment | 0: Bearish, 1: Bullish, 2: Neutral | 0: Bearish (neg.), 1: Bullish (pos.) |
| Poem Sentiment | 0: Negative, 1: Positive, 2: No Impact, 3: Mixed | 0: Negative, 1: Positive |
| Reviews Sentiment Analysis | 0: Negative, 1: Positive | 0: Negative, 1: Positive |
| Auditor Sentiment | 0: Negative, 1: Neutral, 2: Negative | 0: Negative, 1: Positive |

# B    ADDITIONAL EXPERIMENTAL RESULTS

## B.1    GENERALIZATION TO NEW DOMAINS WITH WEAK EXPERTS

One natural question regarding the proposed FoE method is what will happen when encountering a previously unseen domains. We note that for FoE to generalize, it needs to have access to experts that are capable of performing well on (some) samples from the new domain.

To evaluate generalization of FoE to new domains, we modify our MMLU experiment with weak experts. This setting is the most suitable to test generalization as the considered (weak) experts are essentially generalist models capable of handling MMLU categories beyond their strongest areas of expertise. We modified our MMLU experiment by explicitly removing a subset of MMLU categories when training the fuser. Specifically, we removed five categories (*i.e.*, high school european history, business ethics, clinical knowledge, medical genetics, and high

`school us history`) out of 57 MMLU categories. Consequently, during the test phase, data from these categories represent unseen tasks or categories. This modification allows us to assess FoE's generalization capabilities to new data or tasks. In Table 12 we compare the performance of FoE on the aforementioned 5 MMLU categories when they are included and excluded when training the fuser. We see that excluding these categories results in only minor performance decrease, thus demonstrating that FoE is able to generalize to new domains. FoE noticeably improves upon the average expert performance and sometimes even approaches the Oracle expert accuracy.

Table 12: Generalization performance of FoE in our MMLU experiment where five MMLU categories are excluded from FoE training, thus becoming unseen data during test time.

| MMLU Category | Include in FoE training | Exclude from FoE training | Avg. Experts | Oracle Expert |
|---|---|---|---|---|
| high school european history | 68.29 | 68.29 | 54.11 | 68.90 |
| business ethics | 45.45 | 47.47 | 42.96 | 51.52 |
| clinical knowledge | 59.47 | 55.30 | 48.84 | 57.58 |
| medical genetics | 68.69 | 64.65 | 52.05 | 69.70 |
| high school us history | 67.49 | 66.50 | 54.98 | 70.94 |

## B.2 ADDITIONAL RESULTS ON FRUGALFOE OF THE CIFAR-100 TASK

The results of FrugalFoE on the CIFAR-100 super-class classification task (using neural networks as the fuser model) are shown in Figure 5.

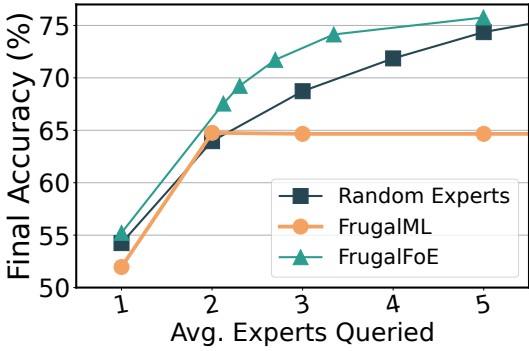

Figure 5: FrugalFoE on CIFAR-100 with neural network as the fuser model.

## B.3 ADDITIONAL EXPERIMENT OF THE SENTIMENT ANALYSIS TASK

The complete results on FrugalFoE of our sentiment analysis task is reported in Table 13.

Table 13: Single expert sentiment analysis results.

| Method | TFN | Poem | Auditor | Reviews Sent. | Avg. |
|---|---|---|---|---|---|
| FoE | 87.54% | 85.71% | 81.71% | 95.20% | **91.88%** |
| TFN Features | 86.93% | 85.71% | 82.32% | 95.20% | 91.81% |
| Poem Features | 87.23% | 82.86% | 81.10% | 95.20% | 91.75% |
| Auditor Features | 86.32% | 82.86% | 82.32% | 95.20% | 91.62% |
| Reviews Sent. Features | 87.23% | 85.71% | 81.10% | 95.20% | **91.88%** |

## B.4 ADDITIONAL EXPERIMENT OF THE SUMMARIZATION TASK

The complete results on FrugalFoE of our summarization task is reported in Table 14.

Table 14: FrugalFoE for the summarization task, ROUGE-2 score (↑) as the evaluation metric.

| Method | CNN DM | XSUM | Multi-News | BillSum | Big-Patent | AESLC | Avg. |
|---|---|---|---|---|---|---|---|
| FoE | 20.2225 | 23.8662 | 18.3499 | 36.9398 | 27.0756 | 20.6736 | **23.7171** |
| CNN DM feature only | 20.1599 | 23.7911 | 18.1450 | 37.8557 | 27.0765 | 20.5473 | **23.7180** |
| XSUM feature only | 20.1402 | 23.8484 | 18.0158 | 36.3483 | 27.0709 | 20.5520 | 23.5963 |
| Multi-News feature only | 19.8900 | 23.3156 | 17.7478 | 18.1870 | 27.0660 | 20.3217 | 21.9752 |
| BillSum feature only | 19.5018 | 23.0579 | 17.5037 | 31.6007 | 27.0413 | 20.3720 | 22.7815 |
| Big-Patent feature only | 19.3697 | 22.6215 | 17.0606 | 37.0621 | 27.0731 | 20.1715 | 22.9850 |
| AESLC feature only | 20.0286 | 23.6388 | 17.8641 | 37.8359 | 27.0661 | 20.3707 | 23.5954 |

## C DETAILS ON THE HYPERPARAMETERS

**Fuser training hyperparameters.** For training the fusers in FoE we use the AdamW optimizer with an initial learning rate at 0.001 and weight decay at $10^{-4}$. We use a batch size of $\{64, 128\}$ across various tasks in our experiments. We train the fuser until convergence in all our experiments, which usually takes $10 - 50$ epochs. We also use the cosine annealing learning rate scheduler for all fuser training.

**Fuser model hyperparameters.** For using neural networks as fusers, we simply use a three-layer MLP as the neural network architecture with ReLU as the activation function. We use a Dropout layer with a dropout rate of 0.5 before the very last fully connected layer in the MLP. The hidden dimensions in our experiments range from 2048 to 8192.

