# OpenReview forum: "Fusing Models with Complementary Expertise"
_ICLR.cc/2024/Conference — ICLR 2024 poster_

### Official Review · Reviewer_w4J8 · 2023-10-27

**Soundness:** 3 good
**Presentation:** 3 good
**Contribution:** 2 fair
**Rating:** 6
**Confidence:** 3

**Summary:**

Acquiring models that generalize across multiple tasks and domains has long been a challenge for the machine learning community. This paper presents the idea of ***Fusion of Experts (FoE)***, aiming to combine the strengths of multiple models with complementary expertise to push their collective generalization capabilities. The main contribution of the paper is to formulate the FoE problem as an instance of supervised learning, which is applicable to both discriminative and generative use cases. In addition, an extended ***FrugalFoE*** has been proposed to allow efficient expert fusion while only evaluating a subset of experts at test time.
Extensive experimental evaluations on a wide range of tasks demonstrate that the proposed fusion method significantly improves the performance of individual experts.

**Strengths:**

- The motivation is sound, and the paper is well written.
- The proposed method casts the FoE problem of fusing outputs of models with complementary expertise as a supervised learning problem. It can be applied to both discriminative and generative use cases.
- The extended Frugal Fusion of Experts (FrugalFoE) allows to efficiently perform expert fusion by only evaluating a subset of experts at test time.
- The proposed fusion method greatly improves the performance of individual experts on a wide range of tasks, while also reducing the number of expert evaluations at test time.

**Weaknesses:**

The primary concern for me is that the proposed fusion method relies on a validation set containing data samples from all $K$ domains, and potentially, the distribution of the validation data is very similar to that of the test data. I can certainly understand that traditional approaches of combining expert predictions may be ineffective, and they mostly use heuristic schemes such as averaging models' outputs or using the most confident model. However, they do not assume that there is additional validation data to access. If there is available validation data, can we train a parameterized fuser for traditional methods? Therefore, this is not fair in a sense, or can I understand that this is a setup for a new fusion task?

Other concerns are as follows.
- In the experimental section, the authors observed that "using a single expert was almost as good as using all experts" on different types of tasks. Could this possibly be the result of an illogical experimental setup? It looks like the knowledge between multiple experts is not complementary but redundant.
- In Table 3, what is the difference between CNN DM Expert (higher part) and CNN DM Expert only (lower part)? Why is there such a wide performance gap between the two?

**Questions:**

- Please take a look at **Weaknesses**.
- Are "Fusion of Experts" and "Mixture of Experts" two different concepts, and what is the essential difference between them?
- How to ensure that the knowledge of different experts is complementary?
- How well does the proposed method perform with test data from outside the $K$ domains?

**Details Of Ethics Concerns:**

No potential ethical issues found.

---

> ### Author Response · Authors · 2023-11-14
> **Responses to reviewer w4J8 (Part-1)**
>
> **Major concern**: *the fusion based method relies on a validation set (+ additional assumption that the validation set should be similar to the test set).*
> **Response**: Please kindly refer to our general response.
> \
> \
> **Concern**: *If there is available validation data, can we train a parameterized fuser for traditional methods? Therefore, this is not fair in a sense, or can I understand that this is a setup for a new fusion task?*
>
> **Response**: We kindly request that the reviewer provides further clarification on their concern, as we are not entirely clear on the issue.
>
> When comparing our proposed FoE method to traditional methods, the key difference lies in the approach. Traditional ensembles aggregate experts' outputs through methods like simple averaging or majority voting. Thus, not like FoE, it is test sample agnostic. The MoE method, on the other hand, requires joint training of experts with the gating mechanism. In contrast, FoE allows the use of pre-developed expert models, which are then fused for enhanced performance. This is achieved by training a lightweight fuser model, which is responsible for selecting the most suitable expert for each test sample.
> \
> \
> **Concern**:  *Using a single expert was almost as good as using all experts: the result of an illogical experimental setup? It looks like the knowledge between multiple experts is not complementary but redundant.*
>
> **Response**: We would like to argue that the experts are indeed complementary, especially when considering Tables 2 and 3, where each expert performs well on only one task, and there is an expert performing well on each task. The term "redundant" might refer to our MMLU experiment, as discussed in the paper. In that experiment, we explored the concept of "weak experts," where none of the LLMs are true experts in any of the MMLU categories. Consequently, it is very likely that their knowledge contains redundancy. However, our experiment shows that even in such a scenario (i.e., beyond our initial problem setting), FoE can still deliver a decent performance boost.
> \
> \
> **Concern**: *Some more explanations about the results in Table 3.*
>
> **Response**: "CNN DM Expert'' refers to the accuracy of the CNN DM expert on each sentiment analysis task, serving as a baseline to indicate individual expert performance. "CNN DM Only" assesses the fuser's performance when only the embedding feature of the CNN DM expert is used for training. This demonstrates that FoE is inherently frugal on the summarization task, as relying on a single expert's embedding is sufficient for accurate expert selection. And that explains why there is a large performance gap between those results. We modified Table 3 to make this less confusing, thanks for pointing it out!
> \
> \
> **Concern**: *Are "Fusion of Experts" and "Mixture of Experts" two different concepts, and what is the essential difference between them?*
>
> **Response**: Let's discuss this under three conditions:
>
> First, when all experts are trained jointly with the gating network in MoE, it results in a new model architecture. Training MoE, such as a SwitchTransformer, can be computationally and resource-intensive. In contrast, FoE focuses on utilizing any pre-trained experts and fusing them, requiring only a small fuser model to be trained. Thus, the development cost of FoE can be much lower than MoE in this scenario.
>
> Second, when all experts are pre-trained, and one aims to build a MoE using these experts with a gating network that directly processes the data (similar to what the reviewer 2UmK asked about), the gating network must be complex, such as a convolutional neural network or ViT for images, or a BERT/RoBERTa model for sequences of tokens. Compared to such scenarios, FoE requires a much simpler fuser architecture, needing nothing more complex than an MLP or kNN model.
>
> Third, when all experts are pre-trained, and one wants to build a MoE with the gating network using experts' outputs or intermediate results, FoE is conceptually identical to MoE in this scenario, however we are unaware of prior work considering MoE in such a setting, nor extending it to the frugal setting as in our paper.

---

> > ### Author Response · Authors · 2023-11-14
> > **Responses to reviewer w4J8 (Part-2)**
> >
> > **Concern**: *How to ensure that the knowledge of different experts is complementary?*
> >
> > **Response**: Examining performance in Table 2 and Table 3, we see that each expert performs well on only one task, and there is an expert performing well on each task. Thus, it’s clear that the experts have complementary expertise. This is easy to check by re-using data for training/validating experts as was done in our experiments.
> > \
> > \
> > **Concern**: *How well does the proposed method perform with test data from outside the $K$ domains?*
> >
> > **Response**:  As discussed in our general response, this issue falls outside the scope of the FoE setting. However, we plan to investigate how FoE will perform if some MMLU categories are excluded during the fuser construction stage and then included in the test stage. We will share the experimental results as soon as they are available.

---

> > > ### Comment · Reviewer_w4J8 · 2023-11-21
> > > **Thanks for response**
> > >
> > > I thank the authors for their thorough and patient responses, and their explanations addressed most of my concerns. After carefully reading the comments of other reviewers and the newly updated paper, I still have some questions.
> > > - To me, the only difference between FoE with one expert only and that expert individually (*e.g.*, **FoE w/ CNN DM Expert features only** vs. **CNN DM Expert** in Table 3) is that FoE additionally trains a *fuser* or *projector* (MLP) on top of the features provided by the expert (**CNN DM Expert**) using validation data. Do I understand correctly?
> > > - If my understanding of the above question is valid, does it mean that we can achieve a decent performance on test data from all domains by simply hooking up a *projector* (MLP) after any expert's feature layer and fine-tuning it with validation data?

---

> > > > ### Author Response · Authors · 2023-11-21
> > > > **Further responses to Reviewer w4J8**
> > > >
> > > > We thank the reviewer for joining the discussion!
> > > >
> > > >
> > > > We are actually not training a projection MLP layer; rather, we are training an MLP to decide which expert to use. So when using **FoE w/ CNN DM Expert features only** at test time, we (1) get features from the CNN DM expert, (2) pass them through the MLP fuser, which predicts the index of the expert to use, and (3) produce the final generation with the predicted expert. **Importantly, this expert does not have to be the CNN DM expert**; it could be the XSUM expert or any other expert LLM (though it could also be the CNN DM expert). On the other hand, the **CNN DM expert** in Table 3 **always** produces generations with this specific LLM.
> > > >
> > > >
> > > > We hope this further clarifies our proposed method. Please let us know if you have any other questions!

---

> > > > > ### Comment · Reviewer_w4J8 · 2023-11-22
> > > > > **One more question**
> > > > >
> > > > > I appreciate the clear explanation, I now better understand the role that the MLP fuser plays. Therefore, the **FoE w/ CNN DM Expert features only** just implies that only **CNN DM Expert's** features are used for training the fuser, and for testing, it is still necessary to send all the experts' features to the fuser to decide which expert's features to use, is that right?

---

> > > > > > ### Author Response · Authors · 2023-11-22
> > > > > > **Our answer to the question**
> > > > > >
> > > > > > **FoE w/ CNN DM Expert features only** implies that only the CNN DM Expert's features are used for training the fuser. For testing, it is **also only** necessary to send the CNN DM Expert’s features to the fuser.
> > > > > >
> > > > > >
> > > > > > Let’s consider a more concrete example: suppose the CNN DM Expert’s embedding dimension is 768, and there are six experts. In this case, the fuser (e.g., an MLP) takes only a 768-dimensional input to perform a six-class classification, that is, to determine which expert to query for a given test data sample. In **FoE w/ CNN DM Expert features only**, we train the MLP using only the CNN DM Expert's 768-dimensional features. During testing, we also use **only** the CNN DM Expert’s features to predict which (out of the six) expert to query.
> > > > > >
> > > > > >
> > > > > > In contrast, in the **regular FoE**, the fuser takes a 6x768=4608-dimensional input to perform the same six-class classification task. During test time, all experts need to be queried for each test data sample to construct the 4608-dimensional input.
> > > > > >
> > > > > >
> > > > > > This is precisely why we claim that FoE is **naturally frugal** for the summarization task, as using only one expert during the entire fusion process (both fuser training and using the trained fuser during testing) is sufficient to obtain good results.

---

> > > > > > > ### Comment · Reviewer_w4J8 · 2023-11-22
> > > > > > > **Thank you for the reply**
> > > > > > >
> > > > > > > Thanks so much to the authors for the careful explanation. All my concerns have been addressed, and I am willing to raise my score to 6.

---

> > > > > > > > ### Author Response · Authors · 2023-11-22
> > > > > > > > **Thank you**
> > > > > > > >
> > > > > > > > Thank you for actively participating in the discussion and for raising the score!

---

### Official Review · Reviewer_HyG2 · 2023-10-27

**Soundness:** 3 good
**Presentation:** 3 good
**Contribution:** 3 good
**Rating:** 8
**Confidence:** 4

**Summary:**

This paper proposes a practical approach to fuse outputs of a set of models that are experts for complementary tasks. Two approaches are proposed:
* FoE-classification, where a fuser is trained on top of the concatenation of outputs.
* FoE-generation, where the fuser learns the optimal choice of expert via cross-entropy.

Additionally, FrugalFoE is proposed, as a strategy to incrementally increase the queries until a cost (loss) criteria is matched. This approach reduces the number of experts being ran.

The experimental section is solid and shows the validity of the approach in various settings, ranging from classification, sentiment analysis, summarization, QA and text generation. Models like ResNet-18 and a wide plethora of LLMs (~7B params) are used, making this evaluation relevant in the state-of-the-art.

**Strengths:**

* The approach is relevant given the availability of pre-trained models nowadays. Methods that smartly fuse models can have impact, since one does not need to re-train, and can incorporate new knowledge to previously trained models.

* The experimental section is very complete, and explores several domains of interest. I specially call out the experiments with LLMs.

* The mathematical derivations are sound.

* The paper is written with clear language, and very few typos.

**Weaknesses:**

* I found the explanation of FrugalFoE harder to follow than the rest. See questions below.

**Questions:**

* To train using the loss in Eq 3.3, one needs to know a priori the labels, ie. which is the correct model for that input. How reasonable is that assumption? Do we also know "exactly" which model should be selected?

* I have some doubts about Equations 4.3 and 4.4 that I would like the authors to clarify. As far as I understand, to obtain the optimal (argmin), we must execute all the experts individually (in Eq. 4.3) and all the subsets in $\mathcal{F}\backslash\tilde{\mathcal{S}}$ for Eq. 4.4. This sounds quite intensive, and definitely more intensive than just running $\mathcal{S}$ experts once. I know there is something I am missing here, I kindly ask the authors to bring some clarity in this sense.

* Why is the cost term in Equation 4.1 summed over $f_k\in \mathcal{S}$? I would have expected this sum to be over $f_k\in \tilde{\mathcal{S}}$, otherwise the term becomes constant wrt. the queried experts, right?

* How can we use $c_k$ in practice? Can we use it to model aspects like energy consumption (for running an expert), flops, etc.?

* Can the authors comment on Figure 2? Why is FrugalML performing so poorly? Even much worse than randomly picking the experts?
  * Additionally, it would be interesting to add std bars for the Random Experts (selecting different random subsets of them, specially at lower values of the x axis).

* In Section 5.2, the authors claim `Though sentiment analysis is essentially a classification task, we train the fuser using the generative model strategy 3.3`. I believe this is a typo and should be "using the classification model strategy".

* I enjoyed the discussion in Section 3.3.

* Conversely, I found Section 4.3 (graph) somehow disconnected and not adding to the work. Unless graphs are used in practice in the code (did not check).

* Minor notation consistency comment. The set $\mathcal{S}$ is not defined when it first appears. Furthermore, one can find both  $k\in\mathcal{S}$ and $f_k\in\mathcal{S}$ in the manuscript, which complicates readability.

---

> ### Author Response · Authors · 2023-11-14
> **Responses to reviewer HyG2**
>
> **Concern**: *How reasonable is that assumption that one knows the correct model for each input? Do we also know "exactly" which model should be selected?*
>
> **Response**: Please kindly refer to our general response.
> \
> \
> **Concern**: *Clarifying Equations 4.1, 4.3, and 4.4.*
>
> **Response**: Thank you for bringing up these questions regarding our equations. We have made changes in the main text to make things clearer. Regarding Eq. 4.1, we sum over $k:f_k \in \mathcal{S}$ because, in our notation, $\mathcal{S}$ includes all experts in $\mathcal{\tilde{S}}$ (queried experts) plus some extra expert we want to query in the next step; therefore, it should not be constant when choosing a new expert. Regarding Eq. 4.3 and 4.4, we assume access to the experts’ outputs for all data points in a validation set. For example, if we have a validation set with a hundred data points and five experts, we have a total of $100\times 5 = 500$ predictions/generations in the validation set. In the first equation, we just choose the expert which had the best average performance in the validation set, while in the second we choose a new expert which is expected to reduce the loss by the biggest amount. The intensive part here is training many fuser models to combine experts in different ways. Still, that cost can be alleviated using k-nearest neighbors as explained in the paragraph “Obtaining fusers for subsets of experts” (Section 4.2).
> \
> \
> **Concern**: *Using $c_k$ in practice.*
>
> **Response**: The $c_k$’s can include different types of costs. It could include energy consumption if experts are run locally or API costs when a third party provides experts. We made this point more explicit in the text.
> \
> \
> **Concern**: *More explanation on why FrugalML performed so poorly, and adding error bars for the random expert baseline.*
>
> **Response**: FrugalML considers composing up to two experts. Thus, in our setting, its performance saturates after two experts (i.e., allowing FrugalML to query more experts is not going to improve its performance). However, when querying either one or two experts, FrugalML still outperforms the approach of selecting random experts. We updated Figure 2 to include error bars for the random expert baseline.
> \
> \
> **Concern**: *Though sentiment analysis is essentially a classification task, we train the fuser using the generative model strategy 3.3. I believe this is a typo and should be "using the classification model strategy".*
>
> **Response**: No, it’s not a typo. The key difference between our generative model strategy and the classification strategy lies in their approaches: while the generative model strategy is designed to predict "which expert to use," the classification strategy directly employs the fuser for classification tasks. In our experiment, even though sentiment analysis inherently involves prediction, we train the fuser to determine which expert should be used for the sentiment analysis task.
> \
> \
> **Concern**: *Notation inconsistency.*
>
> **Response**: Thank you for pointing this out. We have corrected this issue in the main text.

---

> ### Comment · Reviewer_HyG2 · 2023-11-21
> **Answer to rebuttal**
>
> Dear authors,
>
> Thanks for the detailed and concise answers to my questions. I greatly appreciate the effort you put in the rebuttal, specially running extra experiments that help clarifying the overall proposal.
>
> The rebuttal provided answers my most important questions, and the experiments make the paper stronger in my opinion.
> Given the above, I am willing to upgrade my score.
>
> Congratulations on your great work.

---

> > ### Author Response · Authors · 2023-11-21
> > **Thank you!**
> >
> > Our pleasure! Thank you for providing insightful reviews and constructive comments, and for upgrading the score!

---

### Official Review · Reviewer_75JM · 2023-10-30

**Soundness:** 3 good
**Presentation:** 2 fair
**Contribution:** 2 fair
**Rating:** 6
**Confidence:** 4

**Summary:**

This paper introduces an innovative approach for combining various models to optimize performance across different tasks. Recognizing that no single model excels in all tasks and considering the complementary strengths of various pre-trained models, the authors propose a lightweight MLP fuser network trained to either fuse outputs (in the discriminative case) or select the most suitable model (in the generative case) from a pool of $N$ models. To manage the computational expense of querying all $N$ models, they introduce a cost-effective strategy that selects and queries a much smaller subset, $M \ll N$. Their approach, named FrugalFoE, demonstrates impressive results in fusing outputs or selecting expert models, showcasing superior performance in tasks like image classification, summarization, MMLU, and text evaluation.

**Strengths:**

- The paper tackles a novel and significant issue: optimally leveraging different models for diverse tasks. While foundational models generally perform well across various tasks, they have differing strengths. Thus, an approach to effectively combine these models represents a significant advancement.
- The methodology, FrugalFoE, is both technically sound and innovative. It offers a clear problem formulation and further minimizes the need to query all the expert models without sacrificing accuracy.
- The experiments are extensive and cover a range of applications, including image classification, summarization, MMLU, and text evaluation. This demonstrates the method’s versatility and effectiveness.
- The paper effectively situates its work within the broader context of ensemble learning, mixture of experts, and federated learning, thoughtfully introducing these methodologies and their limitations.

**Weaknesses:**

- A primary limitation is the assumption that data for training the fuser are readily available. The proposed approach requires a labeled dataset to train the fuser network. In the discriminative case, we feed the input example to different individual models, take model outputs as the inputs to the fuser network, and train the fuser to predict the label of the input example. Similarly, in the generative case,  we feed the fuser network with individual model outputs and train the fuser to predict the best model index, which is obtained by feeding the labeled data to individual models and selecting the model that achieves the best performance. This prerequisite may not be realistic in practical scenarios, where there is no or only a few labeled data. If a large labeled dataset is available, it might be more efficient to fine-tune a foundational model or employ few-shot learning. A pivotal aspect of this research should be the generalization capabilities of the trained fuser across different tasks (e.g,. train the fuser on some tasks and test the fuser on unseen tasks), which would significantly enhance the paper's contribution.
- The paper lacks an in-depth analysis of the fuser network. Although it is described as a lightweight MLP network, there's no exploration of how different architectures (e.g., simpler networks like linear models or more complex ones like transformers) might impact performance. Additionally, details on the training configurations, such as learning rate, epochs, dataset, and stopping criteria, are missing. The influence of the $K$ parameter in the $K$-NN component of the fuser network is also unclear. What is the rationale for choosing $K=9$?
- The experiment sections lack in-depth descriptions (see questions). The paper would benefit from reallocating less critical sections (like the connection between FrugalFoE and the A* algorithm) to the Appendix and expanding on the experimental details.

**Questions:**

- Page 8, Table 2: Could you clarify the distinction between "TFN Expert Only" and "TFN Expert," and between "Poem Expert" and "Poem Expert Only"?
- Page 8, Table 2: What are the results for the "confidence-based fusion" and "ensemble" baselines? The same question applies to Tables 3 and 4. The confidence-based fusion seems less intuitive in the generative case, how about simply selecting the maximum confidence at each decoding step?
- Page 9, Table 4: Could you explain more details of FoE (Expert 1), FoE (Expert 2), and FoE (Expert 3)? Why does adding more experts appear to degrade performance?
- Page 4: The statement "As long as there is a label shift among the domains, we expect $E[f(X_k)] = E[Y_k]$" needs clarification. Why is this expected?
- Page 6: The phrase "Then $\lambda$ can be interpreted as the final error rate reduction we want to achieve" – could you expound on the reasoning behind this?
- Page 1: The statement "our emphasis is on generalization to test data distributions where none of the experts perform well individually". Shouldn't it be "a few" instead of "none" based on the problem formulation?
- Page 6: The condition "If ... <0 we terminate the search" should be">0"?

---

> ### Author Response · Authors · 2023-11-14
> **Responses to reviewer 75JM (Part-1)**
>
> **Concern**: *assuming data available for training the fuser. Not realistic as there is only a few labeled data. Fine-tune models if labeled data is available? Generalize the trained fuser to other tasks?*
>
> **Response**: Please refer to our general response to this point.
>
> Regarding the suggestion that “If a large labeled dataset is available, it might be more efficient to fine-tune a foundational model or employ few-shot learning,” we contend that in FoE, only a very small fuser model needs to be trained, typically a small MLP or even a kNN, where no training is required. In terms of the number of parameters, the fuser models are usually orders of magnitude smaller than the expert models. This approach is computationally much cheaper than fine-tuning a foundational model, as the reviewer suggested, and also cheaper than fine-tuning any individual expert model.
> \
> \
> **Concern**: *Various choices of the fuser network? More details about the hyper-parameters. Why choose a certain $\kappa$ in the kNN setting?*
>
> **Response**: Thanks for pointing it out! A promising aspect of the FoE is its flexibility in the architecture of the fuser model; it doesn't require any specific design. In our experiments, we explored using both simple MLPs and kNNs as fusers and found that kNNs offer certain advantages, particularly in frugal settings. Additionally, we experimented with simple linear models during the development of the FoE method, although these results are not yet included in the paper. The table below compares the performance on the CIFAR-100 dataset using both MLP and linear models as fusers, each trained on the combined training set of all experts. We observed that using models as complex as MLPs are sufficiently effective, as evidenced by our results in summarization and sentiment analysis tasks where our fuser performs comparably to the oracle models. Therefore, we believe there is no need to explore more complex models like transformers, especially since the input for the fusers is not sequence data, and doing so would compromise the FoE's advantage of having an easily trainable fuser. We will include a discussion on the specific choices of fuser architecture in the revised draft.
>
> | Fuser type      | Final Accuracy (%) |
> | ----------- | ----------- |
> | MLP as the fuser      | **84.23**       |
> | Linear model (logistic regression) as the fuser   | 83.12        |
>
> For questions regarding experimental details, we will revise the experiment section to include more comprehensive descriptions of our setups. The details of our hyper-parameters are already specified in Appendix C. The fuser training process is so simple that there aren’t many hyper-parameters to report.
>
> As for the different choices of $\kappa$, our approach is to select a relatively small and odd number, as is common practice in such experiments. We will update the manuscript with a plot showing results for various values of $\kappa$ shortly, as these experiments are currently in progress. For additional experimental details, we have provided the code used in our experiments, which contains all the necessary information to replicate our results.
> \
> \
> **Concern**: *Expanding experimental details?*
>
> **Response**: We will expand our experiment section in the final version.

---

> > ### Author Response · Authors · 2023-11-14
> > **Responses to reviewer 75JM (Part-2)**
> >
> > **Concern**: *A few detailed questions.*
> > - *The distinction between "TFN Expert Only" and "TFN Expert," and between "Poem Expert" and "Poem Expert Only"?*
> >   **Response**:  "TFN Expert" refers to the accuracy of the TFN expert on each sentiment analysis task, serving as a baseline for individual expert performance. "TFN Expert Only" evaluates the fuser's performance when it is trained using only the embedding feature of the TFN expert. This indicates that FoE is inherently efficient for sentiment analysis tasks since using the embedding of a single expert is enough to accurately predict expert selection. We modified the names in the tables to make them easier to understand.
> >
> > - *What are the results for the "confidence-based fusion" and "ensemble" baselines? Confidence-based fusion seems less intuitive in the generative case.*
> > **Response**: The “confidence-based fusion” is a simple baseline that uses the maximum softmax scores (MaxSoftmax) of each expert as a measure of their confidence for certain test samples. MaxSoftmax is a popular and commonly used technique in the field of uncertainty quantification, always selecting the expert that returns the highest MaxSoftmax score. For the “ensemble” baseline, we average the logits of all experts for prediction. However, we agree that both “confidence-based fusion” and “ensemble” baselines may not be intuitive or even feasible in the generative case, which is why we excluded them from Table 3 and Table 4.
> >
> > - *More details of FoE (Expert 1), FoE (Expert 2), and FoE (Expert 3)? Why does adding more experts appear to degrade performance?*
> > **Response**: The baselines FoE (Expert 1), FoE (Expert 2), and FoE (Expert 3) are similar to the “Poem Expert” and “Poem Expert Only” baselines, as previously explained. These baselines indicate the performance of FoE when using different individual experts' embeddings as features to train the fuser model, rather than the performance of using one, two, or three experts. We have not yet experimented with adding more experts. Our FrugalFoE can effectively identify and eliminate redundant experts, selecting the one that provides the most effective embedding in this weak expert setup. We modified the names in the tables to make them easier to understand.
> >
> > - *Clarification on statement "As long as there is a label shift among the domains, we expect, $E[f(X_k)] = E[Y_k]$”. Why is this expected?*
> > **Response**:  The expression $E[f(X_k)] = E[Y_k]$ simply states that average predictions of an expert across the samples from its domain are approximately equal to class marginals on this domain. This is always true for any reasonable classification model. “We expect” refers to these marginals varying across domains (the last three words in this sentence) in the presence of label shift, which is true by definition of the label shift.
> >
> > - *“Our emphasis is on generalization to test data distributions where none of the experts perform well individually”. Shouldn't it be "a few" instead of "none" based on the problem formulation?*
> > **Response**: No, "none" is correct. Recall that test distribution is a mixture of all domains. Individual experts only perform well on their respective fraction of the test data, i.e., $\frac{1}{K}$ of the test set, thus none of them perform well on the complete test data distribution. This is also evident in our experiments, i.e., in Table 3 individual experts perform poorly on the test distribution as reported in the column "Avg."
> >
> > - *The condition "If ... <0 we terminate the search" should be">0"?*
> > **Response**: Yes, you are right. Thank you for pointing this out! We have fixed the condition.

---

> ### Author Response · Authors · 2023-11-21
> **A follow-up response to Reviewer 75JM**
>
> We would like to thank the reviewer again for the detailed reviews and constructive comments on our paper. As promised, we have updated the manuscript with a plot showing results for various values of $\kappa$. Please refer to the new Figure 2 in the revised manuscript, where one can clearly observe that the performance of FoE does not appear to be sensitive to various values of $\kappa$.
>
> Please let us know if our response has addressed your concerns. Thank you again!

---

> > ### Comment · Reviewer_75JM · 2023-11-22
> >
> > I thank the reviewers for providing a detailed response. As some of my concerns are (partially) addressed, specifically, the generalization of the fuser network to unseen tasks (W1) and the design choice of the fuser network (W2), I'm increasing my rating to 6.
> >
> > However, I still have reservations about their proposed advantages of the fuser network for W1. They mention that the fuser network is better because 1) it can be learned with a few examples, but few-shot in-context learning can also achieve this; 2) it can be trained more efficiently since it has fewer parameters, but many parameter-efficient training methods exist.
> >
> > For W2, it is interesting to learn that MLP outperforms the linear fuser network. However, it is still unclear how other complex architectures, or just MLP with more layers, could improve the task. It is also a bit weird that the authors ablate K=7,9,11 in KNN instead of more general Ks.
> >
> > Further question: could you explain more about what the "FoE w/ TFN Expert Features Only" is? How can you train the fuser with only one expert and apply the fuser to multiple experts?

---

> ### Author Response · Authors · 2023-11-22
> **Further responses to Reviewer 75JM**
>
> Thank you so much for joining the discussion and for increasing your rating! We discuss your remaining questions below.
>
> > They mention that the fuser network is better because 1) it can be learned with a few examples, but few-shot in-context learning can also achieve this; 2) it can be trained more efficiently since it has fewer parameters, but many parameter-efficient training methods exist.
>
> An important aspect of FoE is the ability to reuse pre-trained expert models. It is not clear how to use PEFT to fuse multiple pre-trained LLM experts. One naive approach could be to fine-tune a single expert (or some base model) on all experts’ training data using PEFT. This approach would require fine-tuning an LLM, even with PEFT this could be a fairly expensive task in comparison to training an MLP model as in FoE. For instance, in our experiments, we simply downloaded expert models already available on HuggingFace and did not need to fine-tune any LLM. Meanwhile, fine-tuning an LLM on a combination of experts' domains does not allow one to reuse existing experts and is likely to also require a larger base model to perform well on every domain simultaneously with a single LLM. We believe combining pre-trained experts is more appealing as it can take advantage of the many fine-tuned models open-source community releases.
>
> Regarding in-context learning, the performance is typically worse than that of an expert model. Prior work has demonstrated that training a smaller expert model is typically more efficient than prompting even the largest general model (we provide some references in the first paragraph of the introduction). We also note that the “weak” experts considered in our MMLU experiment are 5-shot prompted general models (which is the standard practice for evaluating on MMLU) and our results demonstrate that it is also beneficial to fuse such ICL-prompted general LLMs.
>
> > However, it is still unclear how other complex architectures, or just MLP with more layers, could improve the task. It is also a bit weird that the authors ablate K=7,9,11 in KNN instead of more general Ks.
>
> We are willing to explore more complex fuser model architectures and will demonstrate the results in the final paper draft. However, we have already demonstrated that FoE is not overly sensitive to the fuser’s model architecture, and even simple MLPs and linear models work sufficiently well. We will also explore more $\kappa$ values in our final paper draft, but we believe it is evident that FoE is reasonably robust to the choice of $\kappa$.
>
> > Could you explain more about what the "FoE w/ TFN Expert Features Only" is? How can you train the fuser with only one expert and apply the fuser to multiple experts?
>
> FoE w/ TFN Expert features only implies that only the TFN Expert's features are used for training the fuser. For testing, it is also necessary to use only the TFN Expert’s features. Let’s consider a concrete example: suppose the TFN Expert’s embedding dimension is 768, and there are six experts. In this case, the fuser (e.g., an MLP) only takes a 768-dimensional input to perform a six-class classification, i.e., to determine which expert to query for a given test data sample. In FoE w/ TFN Expert features only, we train the MLP using only the TFN Expert's 768-dimensional features. During testing, we also use only the TFN Expert’s features to predict which expert to query. In contrast, in the regular FoE, the fuser takes a 6x768=4608-dimensional input for the same six-class classification task. During test time, all experts need to be queried for each test data sample to construct the 4608-dimensional input. This is precisely why we claim that FoE is naturally frugal for the sentiment analysis and summarization tasks, as using only one expert during the entire fusion process (both in training the fuser and using the trained fuser during testing) is sufficient to achieve good results.

---

### Official Review · Reviewer_2UmK · 2023-11-02

**Soundness:** 2 fair
**Presentation:** 3 good
**Contribution:** 2 fair
**Rating:** 6
**Confidence:** 3

**Summary:**

This paper proposes a method to train a set of experts by training the expert one at a time to be fused. Each expert is designed to complement each other during the training procedure by solving the residual gains upon introducing the new expert. Authors evaluated the method on various text domains -- classification, summarization, QA, and generation quality evaluations.

---------

Post rebuttal: With the discussion in the thread, the evaluation has been updated (see other comment for the details).

**Strengths:**

* The proposed method provides an interesting direction that can train multiple models sequentially to train on the residuals of the previous mixture of experts.
* Compared to pure residual approaches, because the transformation function is taken on top of each model’s outputs, we can expect this may be more general than the pure residual learning setting.

**Weaknesses:**

* The proposed algorithm requires multiple experts to be used together, unlike Sparse MoE, which means that the inference cost is multiple times that of each expert. Therefore, the correct baseline for the proposed algorithm is to compare it to an equal number of parameters with the sum of all individual experts. The authors should make this comparison in their paper.
* Similar to the first point, authors provide experimental results on a variety of datasets, however, they did not include many common baselines for each dataset. For example, Figure 2 has a very crude baseline (random experts) or FrugalML/FrugalFoE. I suggest authors to consider common baselines. Few suggestions are Sparse MoE or just a single model with the similar # of parameters.

**Questions:**

Please look at the weakness section.

---

> ### Author Response · Authors · 2023-11-14
> **Responses to reviewer 2UmK**
>
> **Concern**: *The correct baseline for the proposed algorithm is to compare it to an equal number of parameters with the sum of all individual experts or Sparse MoE*.
>
> **Response**:
> We would like to respectfully disagree with the reviewer’s point. We view FoE as a lightweight paradigm that allows us to easily fuse pre-trained expert models to achieve better performance on a mixture of downstream tasks and domains.
>
> It is not very clear how to directly adopt Sparse MoE in a scenario where $K$ pre-trained experts are provided with the data used to build them. One possible approach is to freeze all pre-trained experts and only train the gating network using the available datasets. In such a case, the gating network would need to directly process data, including images and sequences of tokens, requiring a complex architecture like a BERT, a ViT, or a convolutional neural network. This requirement significantly increases the computational difficulty and model complexity needed to build such a Sparse MoE.
>
> Regarding the suggestion to use a model with the combined number of parameters of all experts, we believe this approach is not quite practical. FoE allows us to leverage the resources created by the entire open-source community, such as trained and fine-tuned expert models or LLMs, to achieve better results. For example, one can easily download ten 13B expert LLMs from Hugging Face with just a few lines of code. However, obtaining and running inference on a 130B expert model, as suggested by the reviewer, is far more challenging. In most cases, such a 130B expert LLM does not even exist, and constructing one poses even greater challenges.

---

> > ### Author Response · Authors · 2023-11-22
> > **A friendly reminder**
> >
> > Dear Reviewer 2UmK,
> >
> > Thank you again for your detailed and insightful reviews! We would like to double-check and see if our responses have addressed your concerns, as the deadline for the rebuttal period is around the corner.
> >
> > If you have any further concerns, please feel free to let us know. We are more than happy to clarify more about our paper and discuss it further with you.

---

### Author Response · Authors · 2023-11-14
**General responses to all reviewers**

We thank all reviewers for their constructive and insightful comments. In this section, we provide general responses to the common concerns shared among multiple reviewers. Additionally, we have addressed the specific concerns of each reviewer individually and have modified the draft accordingly to reflect these responses.
  \
  \
**Concern**: *Assumption on the existence of a validation set as well as the true labels of the validation sets (75JM, HyG2, and w4J8)*.

**Response**:
We would like to emphasize that FoE does not require anything beyond the existing datasets used to build the expert models. In the modern ML development pipeline, it is common to use a training set for training or fine-tuning the models and a validation set for early stopping or model selection. This approach is precisely what we employed in our experiments. For example, in our summarization experiments, we downloaded downstream datasets like CNN DailyMail and XSUM and utilized their validation sets to train our fusers. These are exactly the datasets that were used to develop the summarization experts, i.e., the fine-tuned Pegasus models. Moreover, knowing the datasets used to develop each expert also informs us about the domain in which each expert should perform well. For instance, an expert trained on the CNN DailyMail dataset is expected to perform well on the test samples from CNN DailyMail.

We have also tested the FoE approach by exclusively using the training set to build the experts. This was done to demonstrate the method's efficacy even in the absence of a separate validation set. The table below shows an example of the CIFAR-100 experiment, in which the training set was used to train the fuser. It can be observed that utilizing the training set leads to even better fusing results.
| Dataset used to train fuser      | Final Accuracy (%) |
| ----------- | ----------- |
| Training set      | **84.23**       |
| Validation set  | 82.13        |

  \
  \
**Concern**:  *The generalization capability of the fuser to new tasks and the requirement that the test set should be similar to the validation set (75JM and w4J8).*

**Response**:
We thank the reviewers for bringing this interesting point to the table! However, we argue that this concern falls outside the scope of the FoE problem setting. In our FoE setting, we assume that at test time, there will be at least one expert proficient in the domain of the test data query. More specifically, we consider $K$ domains and $K$ experts during both training and test times of the fuser, with training and test sets containing data exclusively from these $K$ domains.

Generalization to unseen domains would only be possible if a subset of the experts considered during FoE happened to also perform well in the new domain. However, this was not the case in our experiments. For example, in the summarization experiment shown in Table 3, all experts performed poorly outside their respective domains.

The only experiment where it would be meaningful to evaluate generalization to new domains is our “weak expert” experiment on MMLU. In this experiment, the fused LLMs are essentially generalist models that can handle MMLU categories beyond their strongest areas of expertise. We propose to modify our MMLU experiment by explicitly removing a subset of weak experts and their associated MMLU categories, such as abstract algebra and anatomy, from the training of the fuser. As a result, during the test phase, data from these categories will represent unseen tasks or categories. This modification will allow us to assess FoE's generalization capabilities to new data or tasks. We will share the results of this modified experiment as soon as they are available. Please let us know if you have any feedback on the proposed experiment.

---

### Author Response · Authors · 2023-11-21
**A follow-up message to all reviewers**

Dear Reviewers,

We have conducted the additional promised experiments based on your questions:


- Updated Figure 2 presents results for varying $\kappa$ hyperparameter of the kNN FrugalFoE algorithm. New results show that our method is robust to the choice of this hyperparameter.
- We conducted a new experiment demonstrating the ability of FoE to generalize to new domains. Please see the results in Table 12 in Appendix B.1. The results show that FoE generalizes well to unseen MMLU categories.


We ask you to please take our rebuttal, new empirical results, and paper revisions into consideration when evaluating our work. We are looking forward to hearing your feedback and would appreciate the response at your earliest convenience as the end of the discussion phase is approaching.


Best,
Authors of the Submission 4271

---

### Meta-Review · Area_Chair_9sWP · 2023-12-05

**Metareview:**

The authors propose a way to fix multiple expert models given access to a dataset which covers all the domains of the experts. The approach trains experts in sequence, learning the residual of the previous experts in order to have subsequent experts be complementary.

The strength of the paper is novelty of both the setting and approach, and broad coverage of the experiments.

Some weaknesses include assumptions about the setting (is it reasonable to assume data on all domains?) and inference cost matching.

**Justification For Why Not Higher Score:**

The inference cost issues is very real - the FoE is effectively a bigger model with higher cost unlike a MoE. The data issue also constrains the setting slightly.

**Justification For Why Not Lower Score:**

The paper has substantial novelty and broad set of experiments, it's likely above the bar of rejection.

---

### Decision · Program_Chairs · 2024-01-16

Accept (poster)